

# Uncertainty caused by resistances in evapotranspiration

Wen Li Zhao[1, 2], Yu Jiu Xiong[3, 4], Kyaw Tha Paw U[4], Pierre Gentine[2], Baoyu Chen[5], Guo Yu Qiu[1]

[1]School of Environment and Energy, Peking University Shenzhen Graduate School, Peking University, Shenzhen 518055, China
[2]Department of Earth and Environmental Engineering, Columbia University, New York, New York, 10027, USA
[3]School of Civil Engineering, Sun Yat-Sen University, Guangzhou 510275, Guangdong, China
[4]Department of Land, Air and Water Resources, University of California at Davis, Davis 95618, USA
[5]Research Institute of Agricultural Resources and Environment, Jilin Academy of Agricultural Sciences, Key Laboratory of Plant Nutrition and Agro-Environment in Northeast Region, Ministry of Agriculture, Changchun, Jilin, China

*Correspondence to*: Yu Jiu Xiong (xiongyuj@mail.sysu.edu.cn), Guo Yu Qiu (qiugy@pkusz.edu.cn)

**Abstract.** Quantifying the uncertainties induced by resistance parameterization is fundamental to understanding, improving, and developing terrestrial evapotranspiration (ET) models. Using high-density eddy covariance (EC) tower observations in a heterogeneous oasis in Northwest China, this study evaluates the impact of resistances on latent heat flux (LE) estimations, the energy equivalent of ET, by comparing resistance parameterizations with varied complexity under one- and two-source

Penman-Monteith (PM) equations. We then discuss possible solutions for reducing such uncertainties by employing a three-temperature (3T) model, which does not explicitly include resistance-related parameters. The results show that the mean absolute percent error (MAPE) varied from 32% to 39% for the LE estimates from the one- and two-source PM equations. When only surface resistance ($r_s$) was parameterized under the one-source network, then the uncertainty (defined as the difference between MAPEs) dropped to 12%. When both $r_s$ and aerodynamic resistance ($r_a$) were parameterized differently

under the one- and two-source networks, then the uncertainties in the estimates were 11~23%, emphasizing that multiple resistances add uncertainties. Additionally, the 3T model performed better than the PM equations, with MAPE of 19%. The results suggest that 1) although prior calibration of the parameters required in resistance estimations can improve the PM-based LE estimates, resistance parameterization process can generate obvious uncertainties, 2) more complex resistance parameterizations leads to more uncertainty in the LE estimation, and 3) the relatively simple 3T model avoids resistance

parameterization, thus introducing less uncertainty in the LE estimation.

## 1 Introduction

Terrestrial evapotranspiration (ET) includes evaporation from soil and water surfaces and transpiration from vegetation. ET links the water, energy and carbon cycles (Oki and Kanae, 2006; Trenberth et al., 2009; Reichstein et al., 2014; Zhang et al., 2016); therefore, accurate estimation of ET is important for improving our understanding of the water and energy cycles,

water-resource management, and agricultural productivity from local to global scales (Mu et al., 2007; Jung et al., 2010). ET can be observed using weighing lysimeters, Bowen ratio or eddy-covariance (EC) systems, such as in the FLUXNET



network (Baldocchi, 2008, 2014; Beringer et al., 2016), including AmeriFlux (USA), the European flux network, AsiaFlux (Asia), Fluxnet Canada, and OzFlux (Australia and New Zealand). Unfortunately, data are limited in terms of the number of stations, their footprint and the observational duration.

Generally, regional or global ET can be estimated using either a water-balance or energy-balance equation (or via a combination thereof) in addition to empirical or semi-empirical methods that relate ET to observations such as radiation, temperature, and vegetation indices (e.g., Carlson et al. 1995; Wang et al., 2007). Among the numerous methods that have been developed, the big-leaf Penman-Monteith (PM) equation (Monteith, 1965) is considered to adequately represent the ET process and is, by far, one of the most widely used (Allen et al., 1998). However, resistance terms used to describe the combined effects of stressors, such as water stress and stomatal closure, on heat and water flux transfer are required in the PM equation, which is similar to other micrometeorological methods (e.g., Shuttleworth and Wallace, 1985) and residual methods of the energy-balance equation (e.g., TSEB, two-source energy balance model; Norman et al., 1995).

The resistance terms can refer to the canopy resistance and the aerodynamic resistance ($r_a$) in a "one-layer/one-source" model (Wang and Dickinson, 2012), and they can also refer to a more complicated and complex resistance network, including the surface resistance, canopy bulk stomatal resistance, boundary layer resistance, and others, in a "multi-layer/ multi-source" model, such as described in Shuttleworth and Gurney (1990), Norman et al. (1995), Zhang et al. (2016), and Deng et al. (2017). Although $r_a$ can be difficult to determine because its calculation depends on certain parameters that are hard to accurately obtain, e.g., the roughness height, zero-plane displacement, and atmospheric stratification (Brutsaert and Stricker, 1979), the uncertainty from $r_a$ is often neglected.

Directly measuring surface or canopy resistance can also be difficult, and these terms are often estimated and scaled from the leaf stomatal resistance or its inverse, the leaf stomatal conductance ($C_s$), using the leaf area index (LAI) (Wang and Dickinson, 2012; Schymanski and Or, 2017), or estimated from flux-tower-based meteorological observation and the inversed one-source PM equation (e.g., Bernhofer et al., 1996; Wu et al., 2018; Li X. et al., 2019). The leaf-level $C_s$ is commonly measured in controlled conditions in laboratory, which do not necessarily capture spatiotemporal variations of the land surface and atmospheric status. Additionally, the $C_s$ values for plant species under different water and climatic combinations are scarce. Furthermore, canopy-level resistance weighted by the LAI from the leaf-level $C_s$ may not capture the variations in turbulence, light and water throughout the canopy and landscape, thereby leading to conflicting behaviors of model performance. Reports have indicated that vegetation sparsity breaks the so-called big-leaf approximation and therefore at low LAI (e.g., < 2) relatively high uncertainties were expected from single source models, such as typically used in the PM equation (e.g., Farahani and Bausch, 1995; Lafleur and Rouse, 1990). Yet, other studies have reported that big-leaf approaches such as used in the PM equation can successfully estimate ET when LAI covers a wide range of values, such as 0.3 to 4 (Rana et al., 1997; Ortega-Farias et al., 2004). Those conflicting results indicate that in a given model, parameterizations have a great impact on ET estimates and the PM equation under proper assumptions can be used to estimate ET correctly only when the resistances are accurately determined.



As a result, the accuracy of the resistance parameterizations is key to better estimating ET for most methods, especially the ones based on the PM equation (Wang and Dickinson, 2012; Li S. et al., 2013; Ershadi et al., 2014; Zhang et al., 2016; Yan et al., 2018) and for land-surface models (e.g., Swenson and Lawrence, 2014). Although new quantitative methods are proposed, e.g., Medlyn et al. (2011) and references therein, a widely used method to parameterize canopy resistance for the

one-source PM equation is the Jarvis-Stewart equation (Jarvis, 1976; Stewart, 1988), which links the canopy resistance to the leaf-level $C_s$ and environmental variables. Another commonly used method is a linear approach which links the canopy resistance to climatic factors and $r_a$ (Katerji and Perrier, 1983). A common limitation in these two methods is that the related environmental (climatic) variables typically vary in time and space (Stewart, 1988) and thus, the empirical functions may need to be calibrated prior to application.

Due to the fact that the big-leaf assumption in the one-source PM-type models might be suitable only for a dense canopy or a bare soil surface (e.g., Shuttleworth and Wallace, 1985; Rana and Katerji, 1998), a two-source model structure was proposed for sparse or tall canopies. In these models, the energy and water flux exchanges among the soil, vegetation, and the atmosphere are coupled via a network of resistances, either in a series or parallel scheme (e.g., Shuttleworth and Wallace, 1985; Norman et al., 1995; Boulet et al., 2015; Li X. et al., 2019). Although the resistances are separated for soil

and canopy, their parameterization methods are similar to those used in the one-source models and require some specification of relatively empirical canopy and soil evaporation resistances.

Remote sensing (RS) can provide physical constraints on the abovementioned methods, and it represents a cost-effective approach to estimating the ET flux at regional to global scales (Bastiaanssen et al., 1998; Kalma et al., 2008; Li Z. et al., 2009; Wang and Dickinson, 2012; Yang et al., 2015; Zhang et al., 2016; Wei et al., 2017). Notwithstanding, surface or

canopy resistances embody complex processes and are difficult to accurately estimate from RS data because they are controlled by a large number of factors, such as wind speed (Su 2002; Sánchez et al. 2008), vegetation type, biophysics, canopy architecture, and soil texture and soil water availability (Leuning, 1995; Shuttleworth and Gurney, 1990; Katerji et al., 2011; Lehmann et al., 2018). These factors are significantly affected by canopy and landscape heterogeneity, and their parameterization process could generate bias and uncertainty and profoundly influence accurate ET estimations (Matheny et

al., 2014; Li et al., 2015; Kustas et al., 2016; Zhang et al., 2016).

Under such conditions, efforts have been made to improve parameterizations of resistances. Some researchers revised soil surface resistance parameterizations (e.g., Katerji et al., 2011; Xu et al., 2017; Lehmann et al., 2018), while others developed new surface-resistance models (e.g., Leuning et al., 2008; Li et al., 2015; Li Y. et al., 2018, 2019; Li X. et al., 2019). In addition to identifying the uncertainty from resistance parameterizations, the influence of the model structure on

ET estimation has been investigated (e.g., Ershadi et al., 2015; Zhao et al., 2015). To avoid the issue of parameterizing resistances, some methods have been proposed to estimate ET without such parameterization, such as the three-temperature (3T) model (Qiu et al., 2006; Xiong et al., 2015; Wang et al., 2016), the Priestley-Taylor method (Priestley and Taylor, 1972), the triangle or trapezoidal method (Price, 1990; Long and Singh, 2012), and the surface-renewal method (Paw U et al., 1995). By eliminating calculation of resistances or local calibration of resistance parameterization, these methods require





relatively fewer inputs but show comparable accuracy (e.g., Long and Singh, 2012; Ershadi et al., 2014; Zhou et al., 2014; Xiong et al., 2015).

Despite these efforts, ET estimation remains biased and accurate ET estimation requires an in-depth identification of uncertainty sources (Long et al., 2014; Ershadi et al., 2015; Zhang et al., 2016; Yao et al., 2017). Therefore, the objectives of
this study are to 1) evaluate resistance-related uncertainties in ET estimates, by employing resistance parameterizations with different complexity under one- and two-source conditions and 2) discuss possible solutions for reducing such uncertainties, including the use of the 3T model, which does not explicitly include resistance-related parameters. The ultimate goals are to identify the uncertainty sources in resistance parameterizations and explore the improvement and development of remotely sensed ET methods.

**2 Description of ET models**

**2.1 One-source PM equation and its parameterization**

The Penman-Monteith (PM) equation is based on a single big-leaf assumption and the energy budget closure, as follows (Monteith, 1965):

$$L(ET) = \frac{\Delta(R_n - G) + \rho_a C_P \, VPD/r_a}{\Delta + \gamma(1 + r_s/r_a)} \tag{1}$$

where $R_n$ is the net radiation, $G$ is the soil heat flux and canopy storage term, $VPD$ represents the vapor pressure deficit of the air, $\rho_a$ is the mean air density at constant pressure, $C_P$ is the specific heat of the air, $\Delta$ represents the slope of the saturation vapor pressure with respect to temperature, $\gamma$ is the psychrometric constant, $r_s$ is the surface resistance and $r_a$ is the aerodynamic resistance.

The PM equation can be extended to include two or multiple sources depending on the configuration of the resistance
networks with respect to heat or water vapor. In this study, ET was estimated by the one-source PM equation (Eq. (1)) and resistances were calculated using two classical methods, i.e., the Jarvis (JA) (1976) and Katerji and Perrier (KP) (1983) methods. In addition, a modified two-source PM equation, proposed for RS applications (Mu et al. 2011), was also used to estimate ET (see section 2.2).

**2.1.1 Parameterizing surface resistance with the JA method**

The classical JA resistance model uses a minimum stomatal resistance term ($r_{smin}$) and environmental factors, such as solar radiation ($R_s$), vapor pressure deficit ($VPD$), air temperature ($T_a$), and soil water ($\theta$), to estimate the canopy resistance, and the LAI is then used to scale the resistance to the entire canopy:

$$r_s = \frac{r_{smin}}{LAI} \frac{1}{(f(R_s)f(VPD)f(T_a)f(\theta))} \tag{2}$$





in which

$$f(R_s) = \frac{R_s(1000 + k_1)}{1000(R_s + k_1)}$$

$$f(VPD) = 1 - k_2 VPD$$

$$f(T_a) = \frac{(T_a - T_L)(T_H - T_a)^t}{(T_{op} - T_L)(T_H - T_{op})^t} \quad where \quad t = \frac{T_H - T_{op}}{T_{op} - T_L} \tag{3}$$

$$f(\theta) = \frac{\theta - \theta_w}{\theta_f - \theta_w}$$

where $r_{smin}$ is the minimum observed stomatal resistance under optimal conditions, i.e., none of the controlling variables are limiting; $T_L$, $T_{op}$, and $T_H$ are the lower, optimal and upper-air temperature limits of stomatal activity, respectively; $\theta_w$ is the

wilting point, $\theta_f$ is the field capacity; and $k_1$ and $k_2$ are treated as constants (Table 1).

### 2.1.2 Parameterizing surface resistance with the KP method

The KP model uses a climatic resistance term ($r^*$), an aerodynamic resistance term ($r_a$) and two empirical coefficients ($c_1$ =0.85 and $c_2$ =1.83) to estimate the canopy resistance and is expressed as follows:

$$r_s = c_1 r^* + c_2 r_a \tag{4}$$

in which

$$r^* = \frac{\Delta + \gamma}{\Delta \gamma} \times \frac{\rho_a C_P VPD}{(R_n - G)} \tag{5}$$

$$r_a = \frac{\ln((z_r - d)/z_{om})\ln((z_r - d)/z_{oh})}{k^2 u_{zr}} \tag{6}$$

where k is von Karman's constant (k=0.4); $z_r$ is the reference height over the canopy at which the wind speed ($u_{zr}$) is measured; $d$ is the zero-plane displacement height, which is commonly taken as 2/3 of the vegetation height; $z_{0m}$ is the local

surface roughness length for momentum transport (in meters) and is assumed to be 0.13 times the vegetation height; and $z_{0h}$ is the local surface roughness for heat (in meters) and is assumed to be equal to $0.1z_{0m}$ (Brutsaert and Stricker, 1979). Equation (6) was used to estimate $r_a$ in the one-source PM equation in sections 2.1.1 and 2.1.2.

### 2.2 Two-source PM equation and its parameterization

According to Mu et al. (2007, 2011), the total ET is regarded as the summation of canopy transpiration ($E_c$) and soil

evaporation ($E_s$), as follows.

$$L(ET) = LE_s + LE_c \tag{7}$$

A brief introduction of the modified two-source PM equation (hereby abbreviated as PM_Mu) is provided below.




(1) Canopy transpiration

The surface canopy resistance is the inverse of the canopy conductance ($C_c$):

$$r_{s,c} = 1/C_c \tag{8}$$

where

$$C_c = \frac{gl\_sh \times (C_s + g\_cu \times r_{corr})}{gl\_sh + C_s + g\_cu \times r_{corr}} \times LAI$$

$$C_s = c_L \times m(T\min) \times m(VPD) \times r_{corr}$$

$$r_{corr} = \frac{1}{\frac{101300}{P} \times \left(\frac{T_a + 273.15}{293.15}\right)^{1.75}}$$

$$m(T\min) = \begin{cases} 1.0 & T\min \geq T\min\_open \\ \dfrac{T\min - T\min\_close}{T\min\_open - T\min\_close} & T\min\_close < T\min < T\min\_open \\ 0.1 & T\min \leq T\min\_close \end{cases}$$

$$m(VPD) = \begin{cases} 1.0 & VPD \leq VPD\_open \\ \dfrac{VPD\_close - VPD}{VPD\_close - VPD\_open} & VPD\_close < VPD < VPD\_open \\ 0.1 & VPD \geq VPD\_close \end{cases} \tag{9}$$

where $C_s$ is the stomatal conductance (m s$^{-1}$); $gl\_sh$ and $gl\_cu$ are the leaf conductance to sensible heat and the cuticular conductance per unit LAI, respectively and are set to constant values (Table 1) (Mu et al., 2007, 2011); $r_{corr}$, a correction factor, is estimated using the pressure ($P$ in Pa) and air temperature ($T_a$ in °C); $c_L$ is the maximum stomatal conductance per unit LAI (m s$^{-1}$); and $m(Tmin)$ ($m(VPD)$) is a multiplier that limits potential stomatal conductance based on the air temperature ($VPD$). The terms *Tmin_close* and *Tmin_open* (*VPD_close* and *VPD_open*, respectively) are threshold values (Table 1).

The aerodynamic resistance ($r_{a,c}$) is estimated using a parallel resistance paradigm for the resistance to convective heat transfer ($r_{cc}$) and the resistance to radiative heat transfer ($r_{rc}$):

$$r_{a,c} = \frac{r_{cc} \times r_{rc}}{r_{cc} + r_{rc}} \tag{10}$$

where

$$r_{cc} = \frac{1}{gl\_bl}$$

$$r_{rc} = \frac{\rho C_p}{4\sigma T_a^3} \tag{11}$$

The latent heat flux of canopy transpiration ($LE_c$) can be estimated using PM equation applied to the vegetation part only, as follows:



$$LE_c = \frac{\Delta \times R_n \times f_c + \rho_a \times C_P \times VPD \times f_c / r_a}{\Delta + \gamma \left(1 + r_s / r_a\right)} \tag{12}$$

(2) Soil evaporation

The surface resistance of the soil surface is estimated as follows:

$$r_{s,s} = r_{totc} \times r_{corr} \tag{13}$$

where

$$r_{totc} = \begin{cases} rbl_{max} & VPD \leq VPD\_open \\ rbl_{max} - \dfrac{\left(rbl_{max} - rbl_{min}\right)\left(VPD\_close - VPD\right)}{VPD\_close - VPD\_open} & VPD\_close < VPD < VPD\_open \\ rbl_{min} & VPD \geq VPD\_close \end{cases} \tag{14}$$

where $r_{totc}$ is an uncorrected total aerodynamic resistance, which is estimated using combinations of the minimal and maximal stomatal resistances ($rbl_{min}$ and $rbl_{max}$) and VPD thresholds for closing and opening of stomata, $VPD\_close$ and $VPD\_open$, and a lack of water stress, respectively. These thresholds were assumed to be a constant according to the biome type (Table 1). The estimation of the aerodynamic resistance ($r_{a,s}$) is similar to Eq. (10), but $r_{cc}$ should be replaced with $r_{s,s}$ estimated via Eq. (13).

Latent heat flux of soil evaporation ($LE_s$) can be estimated using a modified PM equation corrected for relative humidity ($Rh$) and VPD:

$$LE_s = \frac{\Delta \times \left(R_n \times \left(1 - f_c\right) - G\right) + \rho_a \times C_P \times VPD \times \left(1 - f_c\right) / r_a}{\Delta + \gamma \left(1 + r_s / r_a\right)} \times \left(\frac{Rh}{100}\right)^{VPD/\beta} \tag{15}$$

where $\beta$ is assumed to be 200.

**2.3 The 3T model**

The 3T model, which is derived from the energy balance, was first developed by Qiu (1996) and is loosely related to the three-leaf model of Paw U and Daughtry (1984). A unique characteristic of the 3T model is that the estimation of ET does not explicitly include any resistance parameterizations. A reference surface temperature (a dry surface without evaporation

or transpiration) is used to eliminate latent heat and the surface resistance to water vapor (Qiu et al., 1996, 1998), analogous to a leaf that is covered with an impervious surface cover and preventing transpiration (Paw U and Daughtry, 1984). Using this reference soil surface, combined with the assumption that the aerodynamic resistance for the reference soil is the same as that for other soil surfaces (i.e., homogeneous aerodynamic resistance and air temperature), eliminates the aerodynamic resistance from the equation (Eq. (16)).

$$LE_s = R_{n,s} - G_s - \left(R_{n,sr} - G_{sr}\right) \frac{T_{0s} - T_a}{T_{0sr} - T_a} \qquad \text{soil} \tag{16}$$





where $E_s$ is the soil component of the $ET$, in mm s$^{-1}$; L is the latent heat of vaporization; $R_{n,s}$ and $G_s$ are the net radiation and soil heat flux in W m$^{-2}$, respectively; $T_a$ is the air temperature in K; and $T_{0s}$ in K is the temperature for soil component. $R_{n,sr}$, $G_{sr}$, and $T_{0sr}$ are the net radiation, soil heat flux, and temperature for the reference dry soil, respectively.

5       A similar technique is used to eliminate the aerodynamic resistance in the sensible heat equation, assuming that a reference dry vegetation surface has the same aerodynamic resistance as a non-dry vegetation and that the storage and ground heat flux term for the vegetation is small and approximately equal to a scaled ground heat flux of the reference dry vegetation (Eq. (17)).

$$LE_c = R_{n,c} - R_{n,cr} \frac{T_{0c} - T_a}{T_{0cr} - T_a} \qquad\qquad \text{vegetation} \qquad\qquad (17)$$

where $E_c$ is the vegetation component of the ET; $R_{n,c}$ is the net radiation; $T_{0c}$ is the temperature for vegetation component. 10  $R_{n,cr}$ and $T_{0cr}$ are the net radiation and temperature for the reference dry canopy, respectively. The total latent heat flux equation can then be calculated using Eq. (7)

      This unique feature makes the 3T model different from other ET estimation methods that require resistance, such as the PM equation. In recent years, the 3T model has been extended for RS applications and has been shown to be a relatively simple but accurate method (Xiong & Qiu, 2011, 2014; Xiong et al., 2015; Wang et al., 2016).

15  **3 Methods and data sets**

**3.1 Study area and field observations**

The study area is the Heihe oasis in the middle of the Heihe River Basin (Fig. 1). The oasis contains the Yingke and Daman irrigation districts in Zhangye City, Gansu Province, Northwest China, and it is located at 100° 6′ – 100° 52′ E and 38° 32′ – 39° 24′ N. The climate is cold and arid, with a mean annual air temperature, precipitation, and pan evaporation of 7.3 °C, 20  100–250 mm, and 1200–1800 mm, respectively. The elevations of the oasis and the adjacent Gobi Desert range from 1400 m to 1600 m, and the area is generally flat. Maize, spring wheat, vegetables, orchards, and residential areas are the main land-use types in the oasis (Fig. 1).

      In the summer of 2012, an eco-hydrological experiment, namely the Heihe Watershed Allied Telemetry Experimental Research (HiWATER) project, was launched to address water related scientific questions in the arid inland Heihe River 25  Basin (Li X. et al., 2013). The daily datasets from 17 flux towers collected in a key experimental area of 5.5 km × 5.5 km of the HiWATER project from May to September 2012 were used to assess uncertainties in this study:

      Flux observations from eddy-correlation (EC) systems and automatic weather stations (AWS). A total of 21 EC systems (LI-COR Co., Ltd.), which sampled at a frequency of 10 Hz, were installed in the oasis and the adjacent Gobi and desert areas. Seventeen EC systems were installed in a key experimental area of 5.5 km × 5.5 km inside the Yingke and Daman 30  irrigation districts, one was installed in a wetland, and the other three were installed in desert communities outside the oasis





(Li X. et al., 2013). Among the 17 EC systems inside the oasis, 14 were placed in maize fields, and the other three were installed in a vegetable field (No. 1 in Fig. 1 (c)), a residential area (No. 4 in Fig. 1 (c)) and an orchard (No. 17 in Fig. 1 (c)). Detailed information of these 17 systems is summarized in Table 2. In addition, an AWS (Campbell Co., Ltd.) was installed at each EC site to observe and record meteorological data every 10 min, such as the precipitation, air temperature and

humidity, wind speed and direction, air pressure, net radiation and surface temperature. LE values were processed, quality controlled, and recorded every 30 min by a research group (see Liu et al., 2016 for details). If the energy balance closure rate, i.e., (LE+H)/(Rn-G), was less than 0.8 in this study, the LE from the HiWATER project was forcibly changed to achieve closure using the Bowen ratio according to Twine et al. (2000), Barr et al. (2006), Liu et al. (2011) and Xu et al. (2013).

     The LAI was measured by an LAI-2000 instrument (LI-COR Co., Ltd.) from a vegetated area of 5 m× 5 m close to each

EC system inside the oasis (Qu et al., 2014). Twenty measurements were collected during the HiWATER project, but LAI values were unavailable for most sites on several days. Finally, 16-day LAI measurements, representing the seedling, shooting, heading, filling, and maturity stages of maize, were used in this study. The LAI values for all the maize sites are shown in Fig. 2.

**3.2 Assessing the uncertainty in ET estimates from resistances**

To assess the impact of the resistances on the ET estimation (hereby represented by the latent heat flux (LE), its energy equivalent), we evaluated the influence under one- and two-source networks for the PM equation. By comparing the difference between the LE values (hereby abbreviated as LE_JA and LE _KP) estimated by the one-source PM equation (Eq. (1)) according to the JA resistance method (1976) and the KP method (1983), respectively, uncertainty caused by different surface resistance ($r_s$) parameterization is determined. Uncertainty caused by complexity of the resistance networks can be

regarded as the difference between the one-source PM equation and the two-source PM equation (hereby abbreviated as LE_Mu). Comparison results between the one- and two-source PM equations and the 3T model (hereby abbreviated as LE_3T) can indicate the difference between the models with and without resistance-related parameters. In particular, the RS-based PM_Mu (Mu et al., 2011) and the 3T model use a similar model structure (two-source). To avoid generating additional uncertainties in model input estimations, flux-tower observations, rather than RS estimates, were adopted as model inputs to

assess the uncertainties in this study.

**3.2.1 Resistances and LE estimates with the PM equation and tower observations**

The required net radiation and soil heat flux data in the PM equation were obtained directly from flux tower observations, whereas the VPD, psychrometric constant, and slope of the saturation vapor-pressure curve were indirectly estimated from meteorological observations according to FAO paper 56 (Allen et al., 1998) (examples are shown in Tables 3 and 4). The

fractional vegetation cover ($f_c$), which was estimated using the measured LAI according to Walthall et al. (2004), was used to separate the soil and vegetation components when estimating LE using the RS-based PM_Mu equation. The parameters that were required to estimate the resistances in the equations in section 2.2 were estimated by combining meteorological





observations and constant values based on different biome types from Mu et al. (2007, 2011) (an example is shown in Table 5).

### 3.2.2 LE estimates with the 3T model and tower observations

Reference sites are required when applying the 3T model. Flux observations were used as model inputs, so EC system
number 19 (Shenshawo) (Fig. 1b), which was installed in a desert area, was adopted as the dry reference surface. In addition, LE values were estimated using Eq. (16) because the soil heat flux (G) may not be negligible compared to the net radiation (Rn), with a mean ratio of 16% (G/Rn) for vegetated areas. In this case, the observed net radiation, soil heat flux, air temperature, and land surface temperature from EC system number 19 were used as reference values for the other flux towers, and the inputs of the 3T model were from each observation tower (an example is shown in Table 6).

## 4 Results

### 4.1 Characteristics and validation of the LE estimates

The LE values for different land-use types are summarized in Fig. 3 to provide a direct comparison of the LE estimates from the different methods. The EC system observations showed that the LE values increased from non-vegetated areas, i.e., the residential area (193 W m$^{-2}$), to vegetated areas, i.e., maize fields (334 W m$^{-2}$), vegetable land (336 W m$^{-2}$), and orchard
(382 W m$^{-2}$).

The estimated LE values from the one- and two-source PM equations with different resistances parameterizations exhibited significant differences for a given land type. For vegetable land, the estimated LE values for PM_JA, PM_KP, and PM_Mu were 432, 230, and 422 W m$^{-2}$, respectively, whereas those for the maize field were 422, 216, and 377 W m$^{-2}$, respectively. Furthermore, LE heterogeneity could not be adequately captured by the one- and two-source PM equations.
The standard deviation (SD) of the LE values from the maize field was 192 W m$^{-2}$ according to the observations but was 221, 140, and 110 W m$^{-2}$ for the PM_JA, PM_KP, and PM_Mu, respectively. In addition, LE values for the residential area could not be obtained from the PM_JA and PM_Mu methods. Because parameterizing $r_s$ in the PM_JA method was relatively complex and required more inputs compared to the PM_KP method. Consequently, the LE could not be estimated by the PM_JA method when some observations were missing (see Tables 3 to 5 for examples). Alternatively, the parameterizations
for the PM_Mu method were not suitable for barren or sparsely vegetated areas (Mu et al., 2007; Xiong et al., 2015): if the residential land is treated as a soil surface, the resulting LE would be 801 W m$^{-2}$, representing an overestimation of 630 W m$^{-2}$ compared to the EC observation.

The estimated LE values from the 3T model were much closer to the observations, with values of 114, 308, 309, and 452 W m$^{-2}$ for the residential area, vegetable land, maize fields, and orchard, respectively. Heterogeneity of the estimates
also showed similarly to that of the observations. Taken maize field as an example, SD of the observed LE values was 192 W m$^{-2}$ and it was 195 W m$^{-2}$ for the 3T model.





The estimates from the vegetated sites were compared to the EC system observations to further analyze the uncertainty in the LE estimation. Statistics in Fig. 4 showed that the mean absolute percent error (MAPE) between the LE estimates from the PM methods and observations varied, with values of 31.7, 35.5, and 38.9% for PM_JA, PM_KP and PM_Mu, respectively. However, the paired LE values between the 3T model and the observations were generally close to the 1:1 line

(with a MAPE of 18.6%), especially when comparing with the PM equation-based estimates.

Because the maize exhibited varied plant biophysics and soil moisture content conditions (Table 2), we further evaluated the model performance at different maize sites. The mean absolute difference in LE values between the estimates and the observations was 105, 118, 131, and 60 W m$^{-2}$ for PM_JA, PM_KP, PM_Mu, and the 3T model, respectively. PM_KP performed better than PM_JA in terms of $R^2$, with a value of 0.91 for the former compared to 0.78 for the latter.

However, the MAPE values were generally greater than 30% for the PM_KP and PM_JA (Fig. 5). For the PM_Mu, $R^2$ was relatively low with a mean value of 0.43 and MAPE values were relatively large with a mean value of 37.97%. The two-source PM equation showed relatively poor performance compared to the one-source PM equation, which may be likely due to the two-source PM equation requires over complicated parameterizations. For the 3T model, the mean $R^2$ was 0.88, with maximum and minimum values of 0.93 and 0.85, respectively; meanwhile, the MAPE values varied from 13.88% to 22.08%

with a mean value of 17.30%. In addition, the PM equation requires many meteorological observations, so LE cannot be estimated if a necessary variable is missing for a given time. This explains why the number of estimates from the 3T model was larger than that from the PM equations (Tables 3 to 6 and Fig. 4). These results were based on estimations that used observational data, and the effects of potential errors in input estimations were avoided or reduced, indicating the 3T model performed much better than both the one-and two-source PM equations.

Interestingly, the half-hourly estimates and bias of the 3T model for the residential land-use type was 44.6% compared to the flux observation (Fig. 6). If the reference temperatures changed from the land surface temperature of the 19th EC system to thermal-image-based temperatures (see Wang et al., 2016 for details), the MAPE between the LE estimates and observations would be reduced to 31.0%. Although further tests are required, this result indicated that the 3T model, which is commonly applied to soil areas, vegetated areas and combinations thereof, can estimate ET for residential areas.

As shown in Figs. 2 and 7, the investigation data covered different phenological stages and weather conditions during the 2012 growing season. For example, the daytime (7:00-19:00 GMT+8) solar radiation varied from 0 to 1055 W m$^{-2}$, with a mean value of 464 W m$^{-2}$ and SD of 307 W m$^{-2}$ (Fig. 7a). The mean wind speed was 1.5 m s$^{-1}$, with maximum, minimum, and SD values of 5.8, 0.2, and 1.1 m s$^{-1}$, respectively (Fig. 7b). The temperature was 23.1 °C on average, with maximum, minimum, and SD values of 33.1, 8.4, and 4.6 °C, respectively (Fig. 7c). The validation results at a half-hourly scale, as

shown in the above sections, were recorded under different phenological stages and various atmospheric conditions during the growing season, indicating a meaningful comparison.





### 4.2 Uncertainty in the LE estimates

Obviously, the different performances of the one- and two-source PM equations were caused by the different resistance parameterizations. When both $r_a$ and $r_s$ were parameterized differently (based on the data in Fig. 4), the uncertainties (defined as the difference between MAPEs) in the LE estimates varied by 4% and 7% between PM_Mu and PM_KP and

between PM_Mu and PM_JA, respectively. When only $r_s$ was parameterized differently, the difference in LE was 3% between PM_KP and PM_JA. For the one-source method, the JA resistance parameterizations caused the smallest biases in the LE estimation, whereas the empirical KP model led to the largest uncertainty.

As shown in Figs. 3 and 4, the two-source PM_Mu estimate exhibited the largest errors on average. The relatively large bias of the two-source PM_Mu equation indicated that a two-source model with complex inputs and parameterization may

cause more uncertainty when estimating LE over heterogeneous landscapes and thus, there was no benefit of adding complexity compared to a single source model. The resistance estimation processes become more complex as the number of resistances increases (Li et al., 2015). Therefore, biases can be generated from resistance parameterizations and over-complexification, thereby producing error propagation during ET estimation. In contrast, the inputs of the 3T model were based purely on observations, and no variable required additional estimation. The results in this study indicated that the

additional estimation of model inputs commonly produces greater uncertainty.

In addition, wind speed and other parameters, such as the local surface roughness length for momentum transport, are required to calculate the aerodynamic resistances ($r_a$). In our estimation, a margin of 1 m s$^{-1}$ in wind speed will produce differences of 29.4 s m$^{-1}$ in $r_a$ and 54 W m$^{-2}$ in LE (flux Nos. 6 and 15 in Table 4). Because the results were estimated based on flux tower observations and because regional meteorological data, such as the wind speed, are commonly interpolated

using limited ground observations in most RS applications, the estimated data may not accurately represent the real wind-speed distribution in space, producing uncertainty when estimating $r_a$. Furthermore, the quantification and parameterization of surface resistance is difficult. For example, if the JA resistance method is applied at the residential site (No. 4 at 12:30 GMT+8 on July 10), the value of $r_s$ can reach 288 s m$^{-1}$, creating a LE value of 282 W m$^{-2}$, which was 107 W m$^{-2}$ larger than the EC observation. The resistance-estimation method and its parameterization can be major uncertainty sources in ET

estimates.

### 5 Discussion

#### 5.1 Influence of the resistance parameterization on ET

The results in section 4.2 showed that different methods to estimate resistances, especially surface resistances, can produce significant variations in LE estimates, even when using similar driving inputs, such as the net radiation and soil heat flux. As

shown in Fig. 8, the estimated $r_s$ values according to Jarvis (1976) ($r_s$_JA) and Katerji and Perrier (1983) ($r_s$_KP) varied drastically. The mean value of $r_s$_JA was 34 s m$^{-1}$, with maximum and minimum values of 705 and 9 s m$^{-1}$, respectively,





whereas the mean value of $r_s$_KP was 322 s m$^{-1}$, with maximum and minimum values of 992, and 80 s m$^{-1}$, respectively. The average difference between the $r_s$ estimates was 288 s m$^{-1}$, which created a difference of 194 W m$^{-2}$ in LE. The results further indicated that resistances (especially the surface resistance) that were parameterized from different methods could exhibit large differences and were difficult to accurately quantify.

In fact, uncertainty can arise from both different resistance-estimation methods and the same method when using different parameterizations or assumptions. For example, the resistance parameterizations in Mu et al. (2007) were improved in Mu et al. (2011). In particular, the calculation of the surface resistance for vegetation involved different parameterizations of the canopy conductance ($C_c$) (Table 7), the calculation of the surface resistance featured a $r_{totc}$ value of 107 s m$^{-1}$ in Eq. (13), and the constant $\beta$ in the soil-evaporation estimation was changed from 100 to 200 in the improved algorithm (Eq. (15)).

These modifications created different resistance values and LE estimates (hereby abbreviated as LE_Mu_2007 and LE_Mu_2011) (Fig. 9), yet this parameter tweaking is only locally optimized and might not generalize well. The LE values after modification were 104 W m$^{-2}$ larger than those before modification on average, and the root mean squared error (RMSE) between LE_Mu_2007 and LE_Mu_2011 was 142 W m$^{-2}$ (Fig. 9a). The difference was caused by the improvement in the resistance parameterizations. As shown in Figs. 9b and 9c, the modification of the canopy conductance caused a

difference of 39 s m$^{-1}$ between the estimated canopy resistances, whereas the modification of $r_{totc}$ produced differences of 65 s m$^{-1}$ in the soil surface resistance. If the $\beta$ value remained at 100 in the improved algorithm, the estimated LE values would have had a difference of only 2 W m$^{-2}$ on average. Therefore, the large difference (104 W m$^{-2}$) between LE values before and after modification was mainly caused by differences in the resistance parameterizations.

### 5.2 Uncertainty in canopy resistance estimation

Most methods of estimating canopy resistance are based on the leaf conductance (its inverse is the leaf stomatal resistance) as shown in section 2. Although the leaf conductance (or leaf stomatal resistance) can be measured for a single leaf, representing the measurement is questionable. For example, the minimum stomatal resistance ($r_{smin}$) in the JA model requires measuring under optimal conditions when the controlling variables are not limited. Under such conditions, optimal conditions are difficult to find (or determine). For example, when estimating the canopy resistance of maize at an

experimental station in Northwest China, $r_{smin}$ was optimized as 20 s m$^{-1}$ based on observations and simulations between 2007 and 2008 (Li S. et al., 2013) but was found to be 150 s m$^{-1}$ in another study from the same research group (Li et al., 2016). In this study, $r_{smin}$ was set to 20 s m$^{-1}$; if the value were set to 150 s m$^{-1}$, the mean LE values for maize would have been 283 W m$^{-2}$ smaller than that in Table 3, creating obvious differences in the LE estimation (Fig. 10a).

     In addition to the uncertainty arising from $r_{smin}$ during the ET estimation, similar problems are encountered when

attempting to obtain the lower, optimal and upper air temperature limits (or VPD limits) of stomatal activity shown in Table 1. These threshold values likely depend on the species and life cycles of different plants. However, certain values are typically assumed per plant functional types because of the difficulty in obtaining these parameters for each species (e.g., the optimal temperature). For example, Hatfield et al. (2011) summarized the optimal temperatures for several crops, with values




ranging from 26 °C (wheat) to 37 °C (cotton). Another ET model named ETmonitor set the lower, optimal and upper air temperatures to 5, 35, and 55 °C for C4 plants, such as maize, and 0, 25, and 50 °C for C3 plants (Hu and Jia, 2015). Another study by Li S. et al. (2013) suggested that the optimal temperature was 10 and 30 °C for maize and vineyards, respectively. If the optimal temperature used in this study is changed from 35 °C to 10 °C, $r_s$ increases by an average of 369 s m$^{-1}$ relative to

the values in Table 3 and the f (Ta) will be less than 0.08, producing smaller LE values with a mean difference of 208 W m$^{-2}$ (Fig. 10b). In addition, the optimal temperature can vary temporally because of plant acclimation (Kumarathunge et al., 2019).

Furthermore, the non-uniqueness in the parameterizations for a given resistance algorithm (Table 8) increases the complexity of the estimation and generates large uncertainties. For example, algorithms with a linear relationship between f

(Rs) (or f (VPD)) and Rs (or VPD) (Eq. (3)) were used in this study to estimate the surface resistance in the JA model. However, the relationships between f (Rs) and Rs (Mo, 2004; Hu and Jia, 2015) and between f (VPD) and VPD (Li S. et al., 2013, 2015) can be exponential in other algorithms. As shown in Table 3, f (VPD) was approximately 0.91 and the mean $r_s$ was 29 s m$^{-1}$; however, if the exponential algorithm in Li S. et al. (2013) is adopted (Table 8), $r_s$ will be 13 s m$^{-1}$ and f (VPD) will be 1.1, producing higher LE estimates with a mean difference of 54 W m$^{-2}$ compared to the estimates in Table 3.

Additional data in Fig. 10c further suggested that changing the parameterization of f (VPD) can cause overestimation.

Lastly, the canopy resistance is commonly scaled from leaf resistance using the LAI as shown in this and other studies (e.g., Lindroth, 1985; Irmak et al., 2008; Kumar et al., 2014; Zhang et al., 2016). Not only should the up-scaling process be dealt properly (Bernhofer et al., 1996), but also obtaining accurate LAI values are crucial in this scaling process. In addition, LAI is challenging to accurately estimate from space (Gower et al., 1999; Gitelson et al., 2003; Zhu et al., 2016), especially

over dense canopy (as vegetation indices tend to saturate) and over heterogeneous areas, where the error can exceed 1~4 m$^2$ m$^{-2}$ (Houborg et al., 2015; Huang et al., 2016). As reported in Chang et al. (2018), LE estimates from the LAI-based parameterization will exhibit large bias if the LAI fails to capture the vegetation pattern. In this study, a −25% relative change in the LAI could have little effect on the $r_s$ and LE estimates via PM_JA (Fig. 10d), which may be related to the relatively high LAI values during the growing season (e.g., LAI ~ 3 in Table 3). In contrast, a low LAI value can create very

high resistance estimates according to Eq. (2), which has also been reported in other studies (e.g., Ladekarl et al., 2005). The results indicated that LAI values should be validated and calibrated to reduce possible uncertainty during ET estimation and error propagation when estimating ET with resistance-based algorithms and remotely sensed LAI data to scale canopy resistance.

### 5.3 Possible solution for reducing uncertainty in resistance parameterization

When estimating ET through the PM equation or similar methods, uncertainty is likely to propagate due to resistance parameterization, especially when physical boundaries of resistance are required for the parameterization in areas without (or limited) observations and priori knowledge. As such, threshold values of these boundaries (e.g., shown in Table 1) can only be assumed and defined as uncalibrated, which result in errors in ET estimation as discussed in the above sections. Figure 11





shows an example that large differences become obvious between $r_s$_JA (or $r_s$_KP) and $r_s$_PM, which is calculated by inversing the PM equation and site-specific measurements. If these values can be calibrated with available observed datasets, uncertainty caused by resistance parameterization may be limited, which could be a possible solution for reducing uncertainty caused by resistance terms in the ET estimation.

For instance, we calibrated the threshold values in Table 1 through a least squares method based on the observed LE data and the inversed PM equation. The datasets used in this study were divided into two groups: the first group accounted for approximately 20% of the entire available datasets and was used for calibration and the remaining data in the second group was used to perform the validation. The results in Table 1 show that differences exist between the calibrated and uncalibrated values. Notable differences occur in the two empirical coefficients ($c_1$ and $c_2$) in the KP method because the

margin between the calibrated and uncalibrated $c_1$ ($c_2$) values was 0.46 (1.80). If these calibrated values were used to parameterize the resistance and estimate LE, the performance of the one-source PM equation would be improved (Fig. 12), i.e., the LE estimates based on the calibrated resistance (hereby abbreviated as LE_JA_cal and LE_KP_cal) would be closer to the observations and contain less bias compared to the uncalibrated results shown in Fig. 4. The MAPE was reduced, with values of 28.9 and 16.8% for LE_JA_cal and LE_KP_cal, respectively. Specifically, the accuracy of LE_KP_cal improved

significantly as the MAPE decreased by 18.7%. Because resistance requires more parameterizations, the accuracy of LE_JA_cal reduced, with a margin of 2.9%, and the calibrated PM_Mu even performed a little worse compared to the uncalibrated PM_Mu. These results indicate that additional parameterization can increase uncertainty and bias and that the PM equation can be used to estimate ET correctly only when resistances are accurately parameterized or calibrated.

       However, it should be noted that such calibration processes require additional observation datasets, which may be

impossible to obtain in most applications. Even if data are available to perform the calibration, dividing the data into calibration and validation groups is subjective and affects both the calibration and validation results. For example, we tested using 20%, 50%, and 75% of the consecutive datasets to perform calibration and the remaining datasets to perform validation and found that data covering the first 20% since May could lead to the best validation results as shown in Table 1 and Fig. 12. Moreover, different optimal parameter sets likely occur under a given objective in the calibration, i.e., the

parameters are non-unique. As such, while some optimal parameter sets may improve accuracy of the resistance and model performance, the other optimal sets may lead to a poor performance. This phenomenon was obvious for the optimizing parameters for PM_Mu, in which we compared seven parameter sets and the best performance is shown in Fig. 12c. In addition, it was reported that parameter values for estimating canopy conductance in PM method could not be transferable across different time scales (Wang et al., 2016).

**6 Conclusions**

This study investigated the uncertainty in ET estimation from resistance terms by comparing different resistance parameterizations under one- and two-source PM methods. The results showed that the MAPE varied from 32% to 39% for



the one- and two-source PM equations with different resistance parameterizations. If the parameters were calibrated prior to resistance estimation, then the performance of the one-source PM equation could be improved, with MAPE values from 17% to 29%. When only $r_s$ was parameterized differently under the one-source network, then the uncertainty (defined as the difference between MAPEs) was 12%. When both ra and rs were parameterized differently under the one- and two-source

networks, then the uncertainties in the estimates were 11~23%. It was likely that increasing the number of resistance increased the error because too many unknown parameter requires overcomplicated parameterizations. However, the 3T model without resistance parameterization, performed better than the PM equations when resistance estimates were uncalibrated, with MAPE of 19%. We conclude that 1) different resistance parameterizations in the PM equation can produce obvious uncertainties, 2) more complex resistance parameterizations leads to more uncertainty and bias in the ET

estimation, 3) prior calibration of the physical or empirical parameters required in resistance estimations can improve the performance of the PM equation, and 4) the relatively simple two-source 3T model avoids complex parameterization and introduces less uncertainty into the ET estimation.

## Data availability

Observational data are available from the HiWATER project website (http://westdc.westgis.ac.cn/) upon request. Other data

in this study are freely available for research purposes by contacting the authors.

## Author contributions

YJ Xiong conceived and designed the study; Data Preparation, Estimation and Formal Analysis: WL Zhao and YJ Xiong; Writing—Original Draft Preparation: YJ Xiong and WL Zhao; Writing—Review & Editing: YJ Xiong, WL Zhao, GY Qiu, KT Paw U, P Gentine, and BY Chen.

## 20 Competing interests

The authors declare no conflict of interest.

## Acknowledgements

We gratefully acknowledge the financial support from the National Natural Science Foundation of China (41671416), the National Key Research and Development Program of China (2017YFA0604302, 2017YFE0116500), the "Shenzhen

Engineering Laboratory for Water Desalinization with Renewable Energy", and the China Scholarship Council (No. 201806010242, No. 201606380186). We also thank the Heihe Watershed Applied Telemetry Experimental Research (HiWATER) project for their support with the EC observation data.



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



**Table 1. Thresholds assumed for resistance parameterizations.**

| Resistance method | Variables | Uncalibrated values | | | Calibrated values | | |
|---|---|---|---|---|---|---|---|
| | | orchard | residential | cropland | orchard | residential | cropland |
| Jarvis model (1976) | $r_{smin}$ (s m$^{-1}$) | 20 | 20 | 20 | 25.05 | 25.05 | 25.05 |
| | $k_1$ (W m$^{-2}$) | 5 | 5 | 5 | 5.48 | 5.48 | 5.48 |
| | $k_2$ (hPa$^{-1}$) | 0.023 | 0.023 | 0.023 | 0.033 | 0.033 | 0.033 |
| | $T_L$ (°C) | 5 | 5 | 5 | 7.89 | 7.89 | 7.89 |
| | $T_{op}$ (°C) | 35 | 35 | 35 | 36.96 | 36.96 | 36.96 |
| | $T_H$ (°C) | 55 | 55 | 55 | 53.44 | 53.44 | 53.44 |
| Katerji and Perrier model (1983) | $c_1$ | 0.85 | 0.85 | 0.85 | 0.39 | 0.39 | 0.39 |
| | $c_2$ | 1.83 | 1.83 | 1.83 | 0.03 | 0.03 | 0.03 |
| PM_Mu (Mu et al., 2011) | Tmin_open (°C) | 8.61 | 12.02 | 12.02 | 8.00 | 8.00 | 8.00 |
| | Tmin_close (°C) | -8.00 | -8.00 | -8.00 | -6.00 | -6.00 | -6.00 |
| | VPD_close (Pa) | 4300 | 4200 | 4500 | 4500 | 4500 | 4500 |
| | VPD_open (Pa) | 650 | 650 | 650 | 1000 | 1000 | 1000 |
| | gl_sh (m s$^{-1}$) | 0.04 | 0.02 | 0.02 | 0.012 | 0.012 | 0.012 |
| | $c_L$ (m s$^{-1}$) | 0.0065 | 0.007 | 0.007 | 0.007 | 0.007 | 0.007 |
| | $rbl_{max}$ (s m$^{-1}$) | 20 | 20 | 20 | 22.29 | 22.29 | 22.29 |
| | $rbl_{min}$ (s m$^{-1}$) | 55 | 50 | 50 | 49.77 | 49.77 | 49.77 |

Note: 1. The uncalibrated values were directly cited from the literature, whereas the calibrated values were deduced from the observed LE data in this study and inversed PM equation through a least squares method (see section 5.3 for details).

2. Calibration of the PM_Mu was performed by neglecting the biome difference because only one EC system is applied for orchard and residential areas and dataset availability was limited.



**Table 2. Details of the 17 eddy-covariance (EC) systems in the key experimental area shown in Fig. 1c.**

| Sensor type & manufactures | EC system ID | Sensor height (m) | observation duration | LAI ($m^2\ m^{-2}$) | soil moisture (%) |
|---|---|---|---|---|---|
| Gill/Li7500A, Gill, UK/Li-cor, USA | 1 | 3.8 | Jun. 16 to Sep. 17 | 1.83 | 36.04 |
|  | 3 | 3.8 | Jun. 25 to Sep. 18 | 2.35 | 36.57 |
|  | 9 | 3.9 | Jun. 25 to Sep. 17 | 3.42 | 35.99 |
| Gill/Li7500, Gill, UK/Li-cor, USA | 16 | 4.9 | Jul. 2 to Sep. 17 | 2.61 | 28.76 |
| CSAT3/Li7500A, Campbell/Li-cor, USA | 4 | 4.2 (6.2 after Aug.19) | May 31 to Sep. 17 | – | 18.65 |
|  | 6 | 4.6 | May 28 to Sep. 21 | 2.39 | 31.57 |
|  | 7 | 3.8 | May 29 to Sep. 18 | 2.41 | 29.59 |
|  | 13 | 5 | May 27 to Sep. 20 | 2.27 | 22.62 |
|  | 15 | 4.5 | May 25 to Dec. 30 | 3.14 | 28.45 |
| CSAT3/Li7500, Campbell/Li-cor, USA | 2 | 3.7 | Jun. 15 to Sep. 21 | 2.49 | 20.73 |
|  | 5 | 3 | Jun. 3 to Sep. 18 | 2.50 | 29.83 |
|  | 8 | 3.2 | May 28 to Sep. 21 | 2.58 | 31.53 |
|  | 10 | 4.8 | Jul. 5 to Sep. 17 | 2.62 | 30.14 |
|  | 11 | 3.5 | May 29 to Sep. 18 | 2.26 | 27.12 |
|  | 12 | 3.5 | May 28 to Sep. 21 | 2.44 | 20.82 |
|  | 14 | 4.6 | May 30 to Sep. 17 | 2.53 | 21.83 |
| CSAT3/EC150, Campbell, USA | 17 | 7 | May 31 to Sep. 17 | 1.65 | 28.90 |

Note: 1. All the sensor types are open-path and related information were cited from Liu et al. (2016);

2. Sampling frequency of the EC systems was 10 Hz and the EC data was post-processed, quality controlled, and recorded
every 30 min on average by Liu et al. (2016) and distributed by the HiWATER project;

3. LAI and soil moisture values were averaged using corresponding data provided by the HiWATER project.





**Table 3. Inputs and output of the one-source PM equation based on the JA resistance parameterizations and flux tower observations at 12:30 GMT+8 on July 10, 2012.**

| Flux tower | | Inputs | | | | | | | | | | Output |
|---|---|---|---|---|---|---|---|---|---|---|---|---|
| | | Observation | | | | Estimation based on observation | | | | | | |
| Land use | ID | Rn (W m$^{-2}$) | G (W m$^{-2}$) | LAI (m m$^{-2}$) | VPD (hPa) | $\Delta$ | $f(R_s)$ | $f(VPD)$ | $f(T_a)$ | $f(\theta)$ | $r_s$ (s m$^{-1}$) | $r_a$ (s m$^{-1}$) | LE_JA (W m$^{-2}$) |
| vegetable land | 1 | 656.70 | 34.15 | 1.26 | 25.14 | 0.21 | 1.00 | 0.42 | 0.91 | 0.60 | 68.66 | 38.01 | 578.95 |
| maize fields | 2 | 689.99 | 90.50 | 4.07 | 23.92 | 0.20 | 1.00 | 0.45 | 0.91 | 0.70 | 17.26 | 38.89 | 712.75 |
| | 5 | 715.00 | 38.55 | 3.29 | 24.17 | 0.21 | 1.00 | 0.44 | 0.92 | 0.70 | 21.39 | 31.34 | 800.92 |
| | 6 | 678.30 | 186.73 | 3.15 | 23.97 | 0.20 | 1.00 | 0.45 | 0.91 | 0.70 | 22.22 | 33.57 | 652.87 |
| | 8 | 720.00 | 104.94 | 2.64 | 21.59 | 0.21 | 1.00 | 0.50 | 0.91 | 0.70 | 23.55 | 43.89 | 654.50 |
| | 11 | 714.80 | 57.49 | 2.66 | 23.28 | 0.20 | 1.00 | 0.46 | 0.90 | 0.70 | 25.61 | 30.59 | 764.05 |
| | 12 | 714.12 | 43.53 | 2.60 | 22.95 | 0.20 | 1.00 | 0.47 | 0.91 | 0.70 | 25.53 | 44.06 | 701.26 |
| | 13 | 709.72 | 63.40 | 2.96 | 22.34 | 0.20 | 1.00 | 0.49 | 0.91 | 0.70 | 21.87 | 57.80 | 649.08 |
| | 15 | 680.00 | 39.76 | 2.52 | 19.96 | 0.20 | 1.00 | 0.54 | 0.90 | 0.70 | 23.34 | 62.80 | 613.28 |
| orchard | 17 | 734.10 | 59.73 | 1.70 | 23.31 | 0.21 | 1.00 | 0.46 | 0.92 | 0.65 | 42.40 | 52.42 | 642.28 |

Note: Not all EC systems were available for all measurements, the algorithm may not be suitable for residential land, and some LE_JA could not be estimated due to lack of meteorological observation;

VPD and $\Delta$ were estimated according to FAO paper 56 (Allen et al., 1998);

when parameterizing $r_s$, the minimum resistance for a leaf ($r_{smin}$) was set according to Mu et al. (2011) and Li S. et al. (2013), i.e., $r_{smin}$=20 s m$^{-1}$;

when estimating $r_a$, the height of all vegetation was assumed to be 2 m.



**Table 4. Inputs and output of the one-source PM equation based on the Katerji and Perrier resistance parameterizations and flux tower observations at 12:30 GMT+8 on July 10, 2012.**

| Flux tower | | Inputs | | | | | | | Output |
|---|---|---|---|---|---|---|---|---|---|
| | | Observation | | | Estimation based on observation | | | | |
| Land use | ID | Rn | G | $\Delta$ | VPD | $r^*$ | $r_s$ | $r_a$ | LE_ KP |
| | | (W m$^{-2}$) | (W m$^{-2}$) | | (hPa) | (s m$^{-1}$) | (s m$^{-1}$) | (s m$^{-1}$) | (W m$^{-2}$) |
| vegetable land | 1 | 656.70 | 34.15 | 0.21 | 25.14 | 115.51 | 167.75 | 38.01 | 409.70 |
| residential area | 4 | 567.00 | 81.50 | 0.22 | 27.41 | 165.94 | 201.51 | 33.04 | 349.54 |
| | 2 | 689.99 | 90.50 | 0.20 | 23.92 | 119.10 | 172.40 | 38.89 | 398.07 |
| | 5 | 715.00 | 38.55 | 0.21 | 24.17 | 106.41 | 147.81 | 31.34 | 458.03 |
| | 6 | 678.30 | 186.73 | 0.20 | 23.97 | 145.64 | 185.22 | 33.57 | 341.23 |
| | 8 | 720.00 | 104.94 | 0.21 | 21.59 | 104.46 | 169.11 | 43.89 | 399.90 |
| maize fields | 11 | 714.80 | 57.49 | 0.20 | 23.28 | 106.32 | 146.36 | 30.59 | 442.37 |
| | 12 | 714.12 | 43.53 | 0.20 | 22.95 | 102.02 | 167.34 | 44.06 | 434.63 |
| | 13 | 709.72 | 63.40 | 0.20 | 22.34 | 103.24 | 193.53 | 57.80 | 407.25 |
| | 15 | 680.00 | 39.76 | 0.20 | 19.96 | 93.67 | 194.54 | 62.80 | 394.82 |
| orchard | 17 | 734.10 | 59.73 | 0.21 | 23.31 | 102.56 | 183.10 | 52.42 | 433.05 |

Note: Not all EC systems were available for all measurements and some LE_KP could not be estimated due to lack of meteorological observation;

VPD and $\Delta$ were estimated according to FAO paper 56 (Allen et al., 1998);

when estimating $r_a$, the height of all vegetation was assumed to be 2 m.





**Table 5. Inputs and output of the modified two-source PM equation (Mu et al. 2011) and flux tower observations at 12:30 GMT+8 on July 10, 2012.**

| Flux tower | | Inputs | | | | | | | | | Output |
|---|---|---|---|---|---|---|---|---|---|---|---|
| | | Observation | | | | Estimation based on observation | | | | | |
| Land use | ID | Rn | G | Δ | VPD | $f_c$ | vegetation | | soil | | LE_Mu |
| | | (Wm$^{-2}$) | (Wm$^{-2}$) | | (hPa) | | $r_{s,c}$ (s m$^{-1}$) | $r_{a,c}$ (s m$^{-1}$) | $r_{s,s}$ (s m$^{-1}$) | $r_{a,s}$ (s m$^{-1}$) | (Wm$^{-2}$) |
| vegetable land | 1 | 656.70 | 34.15 | 0.21 | 25.14 | 0.47 | 291.22 | 49.99 | 30.09 | 30.09 | 563.80 |
| | 2 | 689.99 | 90.50 | 0.20 | 23.92 | 0.87 | 89.41 | 49.99 | 27.88 | 27.88 | 477.91 |
| | 5 | 715.00 | 38.55 | 0.21 | 24.17 | 0.81 | 112.08 | 49.99 | 27.94 | 27.94 | 475.70 |
| | 6 | 678.30 | 186.73 | 0.20 | 23.97 | 0.79 | 115.98 | 49.99 | 27.81 | 27.80 | 474.71 |
| | 8 | 720.00 | 104.94 | 0.21 | 21.59 | 0.73 | 125.99 | 49.99 | 26.33 | 26.33 | 463.82 |
| maize fields | 11 | 714.80 | 57.49 | 0.20 | 23.28 | 0.74 | 133.59 | 49.99 | 27.33 | 27.33 | 479.75 |
| | 12 | 714.12 | 43.53 | 0.20 | 22.95 | 0.73 | 134.78 | 49.99 | 27.18 | 27.18 | 477.89 |
| | 13 | 709.72 | 63.40 | 0.20 | 22.34 | 0.77 | 115.48 | 49.99 | 26.80 | 26.80 | 463.43 |
| | 14 | 701.13 | 126.40 | 0.21 | 24.15 | 0.79 | 116.81 | 49.99 | 27.90 | 27.90 | 476.48 |
| | 15 | 680.00 | 39.76 | 0.20 | 19.96 | 0.72 | 124.97 | 49.99 | 25.22 | 25.22 | 452.58 |
| orchard | 17 | 734.10 | 59.73 | 0.21 | 23.31 | 0.57 | 202.29 | 25.00 | 29.91 | 29.91 | 522.65 |

Note: Not all EC systems were available for all measurements, the algorithm may not be suitable for residential land, and some LE_Mu could not be estimated due to lack of meteorological observation;

VPD and Δ were estimated according to FAO paper 56 (Allen et al., 1998).





**Table 6. Inputs and output of the 3T model based on flux tower observations at 12:30 GMT+8 on July 10, 2012.**

| Flux tower | | Inputs | | | | | | | Output |
|---|---|---|---|---|---|---|---|---|---|
| Land use | ID | Rn (W m$^{-2}$) | G (W m$^{-2}$) | Ts (°C) | Ta (°C) | Tsr (°C) | Rnr (W m$^{-2}$) | Gr (W m$^{-2}$) | LE_3T (W m$^{-2}$) |
| vegetable land | 1 | 656.70 | 34.15 | 35.41 | 26.82 | 59.89 | 556.80 | 84.74 | 499.93 |
| residential area | 4 | 567.00 | 81.50 | 49.04 | 28.12 | 59.89 | 556.80 | 84.74 | 174.66 |
| | 2 | 689.99 | 90.50 | 33.66 | 26.41 | 59.89 | 556.80 | 84.74 | 497.28 |
| | 3 | 721.50 | 33.44 | 26.68 | 27.28 | 59.89 | 556.80 | 84.74 | 696.75 |
| | 5 | 715.00 | 38.55 | 25.99 | 26.89 | 59.89 | 556.80 | 84.74 | 689.32 |
| | 6 | 678.30 | 186.73 | 28.33 | 26.62 | 59.89 | 556.80 | 84.74 | 467.31 |
| | 8 | 720.00 | 104.94 | 28.60 | 26.76 | 59.89 | 556.80 | 84.74 | 588.84 |
| | 9 | 717.40 | 65.58 | 26.56 | 27.20 | 59.89 | 556.80 | 84.74 | 661.06 |
| maize fields | 10 | 700.80 | 48.95 | 27.03 | 27.25 | 59.89 | 556.80 | 84.74 | 655.03 |
| | 11 | 714.80 | 57.49 | 28.43 | 26.25 | 59.89 | 556.80 | 84.74 | 626.72 |
| | 12 | 714.12 | 43.53 | 32.66 | 26.71 | 59.89 | 556.80 | 84.74 | 585.94 |
| | 13 | 709.72 | 63.40 | 33.08 | 26.48 | 59.89 | 556.80 | 84.74 | 553.09 |
| | 14 | 701.13 | 126.40 | 31.59 | 26.77 | 59.89 | 556.80 | 84.74 | 506.03 |
| | 15 | 680.00 | 39.76 | 31.00 | 26.08 | 59.89 | 556.80 | 84.74 | 571.55 |
| | 16 | 653.40 | 58.00 | 27.27 | 27.18 | 59.89 | 556.80 | 84.74 | 594.10 |
| orchard | 17 | 734.10 | 59.73 | 26.69 | 27.13 | 59.89 | 556.80 | 84.74 | 680.71 |

Note: The reference parameters, i.e., Tsr, Rnr, and Gr, are observations from EC system number 19 (Shenshawo), which is located in a desert area.

Not all EC systems were available for all measurements.





**Table 7. Differences in the surface resistance estimations between the old and improved algorithms by Mu et al. (2007, 2011)**

| | Differences | |
| --- | --- | --- |
| | Old algorithms in Mu et al. (2007) | Improved algorithms in Mu et al. (2011) |
| Surface resistance for vegetation | $C_c = C_s \times LAI$ | $C_c = \dfrac{gl\_sh \times (C_s + g\_cu \times r_{corr})}{gl\_sh + C_s + g\_cu \times r_{corr}} \times LAI$ |
| Surface resistance for soil | $r_{s,s} = r_{totc} \times r_{corr}$ <br> where $r_{totc}$=107 s m$^{-1}$. | $r_{s,s} = r_{totc} \times r_{corr}$ <br> where $r_{totc}$ is a function of VPD as shown in Eq. (13). |
| Soil evaporation (Eq. (15)) | $\beta$=100 | $\beta$=200 |

5 **Table 8. Different algorithms that represent environmental forces for leaf resistance in the JA model.**

| algorithm | Linear | exponential |
| --- | --- | --- |
| $f(R_s)=$ | $\dfrac{R_s(1000 + k_1)}{1000(R_s + k_1)}$ (Li S. et al., 2013, 2015) | $1 - \exp\left(-\dfrac{R_s}{500}\right)$ (Mo, 2004; Hu and Jia, 2015) |
| $f(T_a)$ | $t = \dfrac{T_H - T_{op}}{T_{op} - T_L}$ (Stewart, 1988; Hu and Jia, 2015) | $t = \dfrac{T_{op} \times T_H}{T_{op} - T_L}$ (Li S. et al., 2013, 2015) |
| $f(VPD)=$ | $1 - k_2 VPD$ (Mo, 2004; Hu and Jia, 2015) | $\exp(-k_2 VPD)$ (Li S. et al., 2013, 2015) |



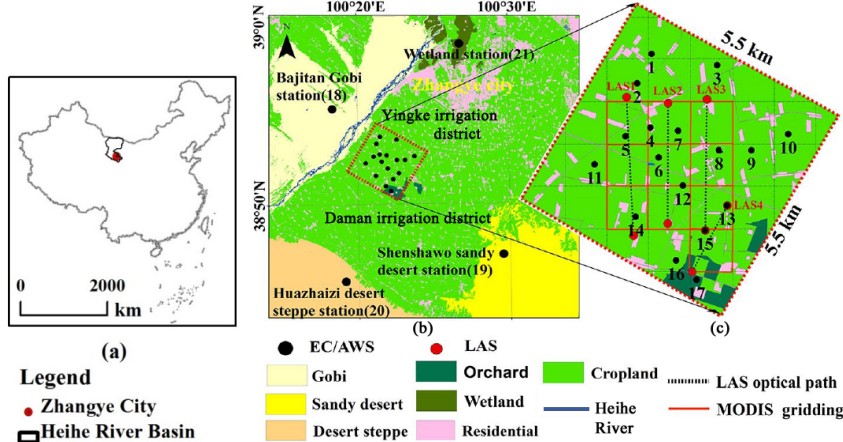

**Figure 1: Study area: (a) location of Zhangye and the Heihe River Basin in China, (b) 21 EC flux towers in HiWATER, and (c) detailed locations of the 17 EC systems inside the key experimental area in the Zhangye Oasis. Figs (b) and (c) were revised from Ma et al. (2015); AWS and LAS are abbreviations for automatic weather station and large-aperture scintillometer, respectively.**

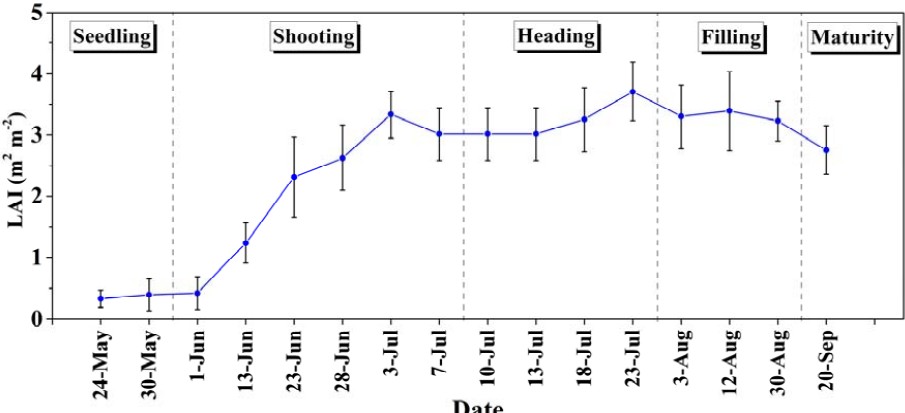

**Figure 2: Variations in the average leaf area index (LAI) from 14 maize fields. The error bars denote the standard deviation of the mean observation.**





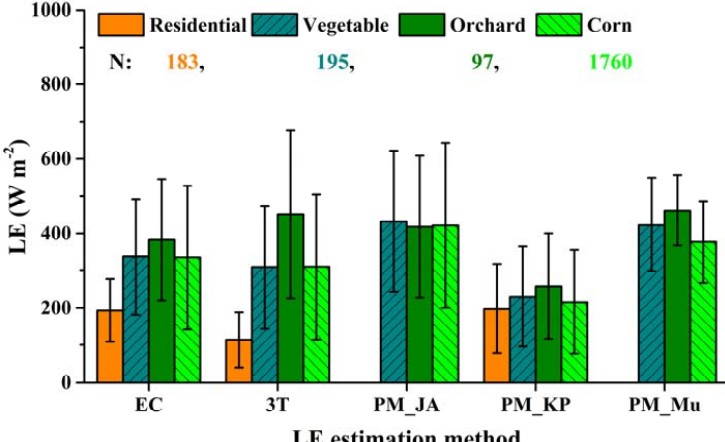

Figure 3: Latent heat flux (LE) values of different land types. The error bars denote the standard deviation of the mean estimate.





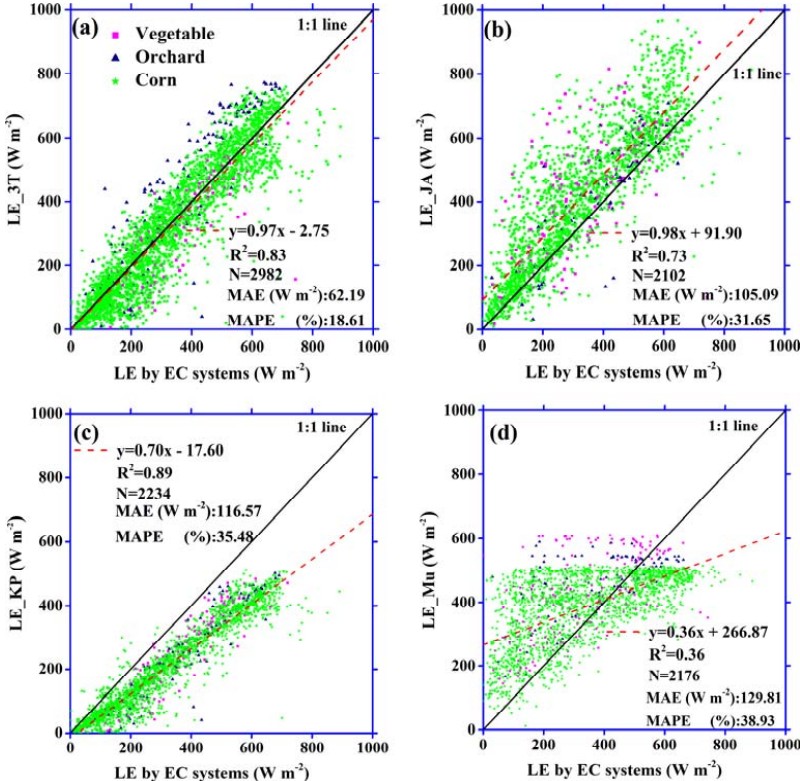

Figure 4: Comparison of the estimated and measured latent heat flux (LE) values at vegetated flux towers. MAE and MAPE represent the mean absolute error and absolute percent error, respectively. The temporal scale of the data is one half hour.



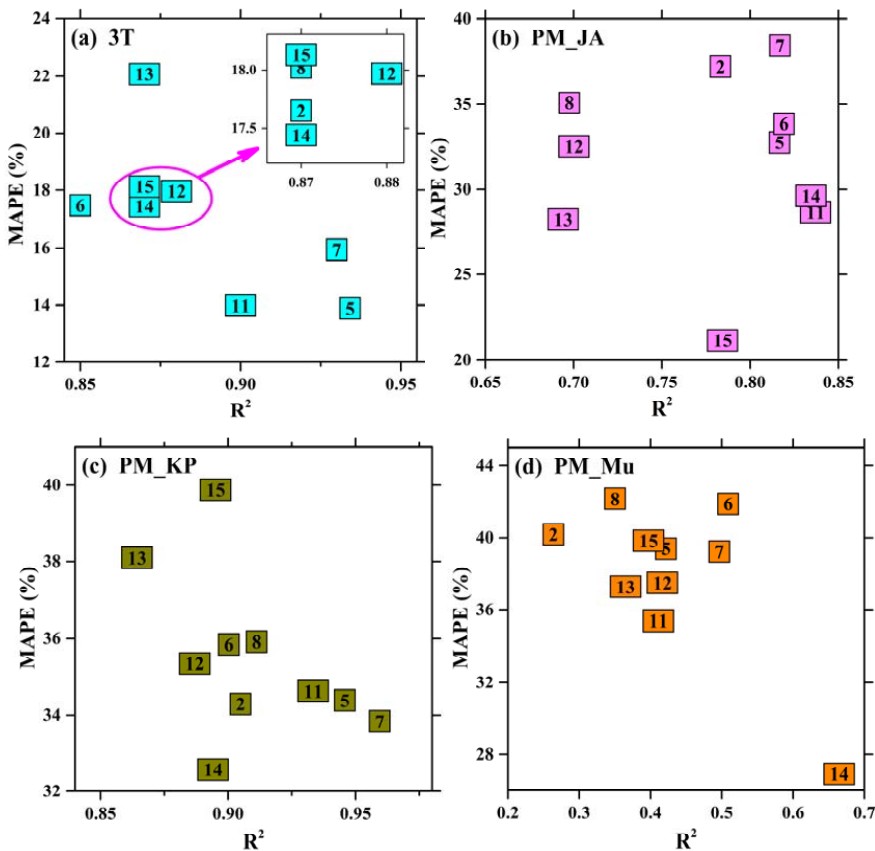

**Figure 5: Performance of different models at different maize sites based on data from Figure 4. The number inside the rectangle represents the EC system ID in Figure 1c.**



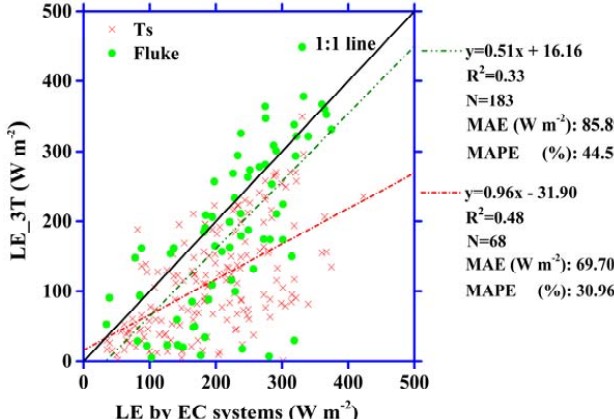

**Figure 6: Comparison of the half-hourly ET estimates from the 3T model and measurements at a residential flux site. MAE and MAPE represent the mean absolute error and absolute percent error, respectively. The red crosses represent the observed land surface temperatures (Ts) at the 19ᵗʰ EC system, which were adopted as reference temperatures when estimating LE by the 3T model, whereas the green dots were estimated using Fluke-thermal-image-based temperatures as reference temperatures.**



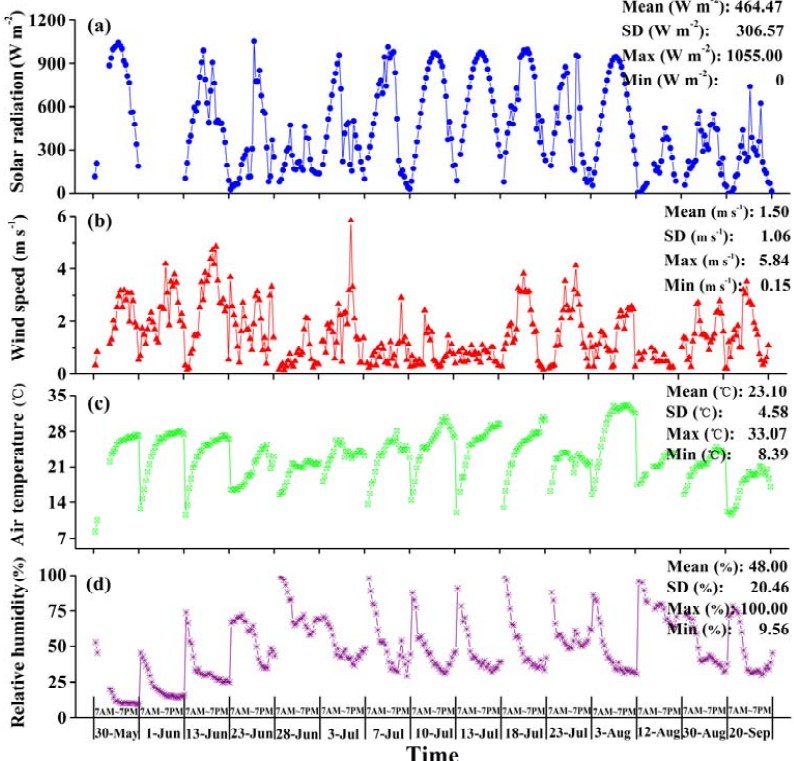

**Figure 7: Variations in the daytime (7:00-19:00 GMT+8) solar radiation (a), wind speed (b), air temperature (c), and relative humidity (d) for the investigated days at the 15$^{th}$ EC tower during the 2012 growing season.**





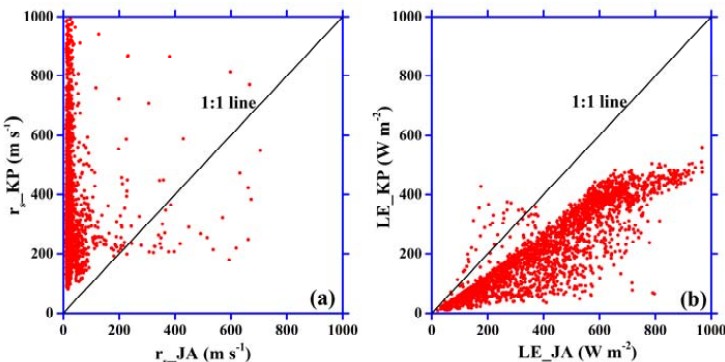

**Figure 8: Differences in latent heat flux (LE) estimates from different $r_s$ calculation methods: (a) comparison of $r_s$ values from the Jarvis (JA) (1976) and Katerji and Perrier (KP) (1983) methods, and (b) the effect on LE estimation.**
**Note: The LE values were estimated using the one-source PM equation and the same inputs except for $r_s$.**




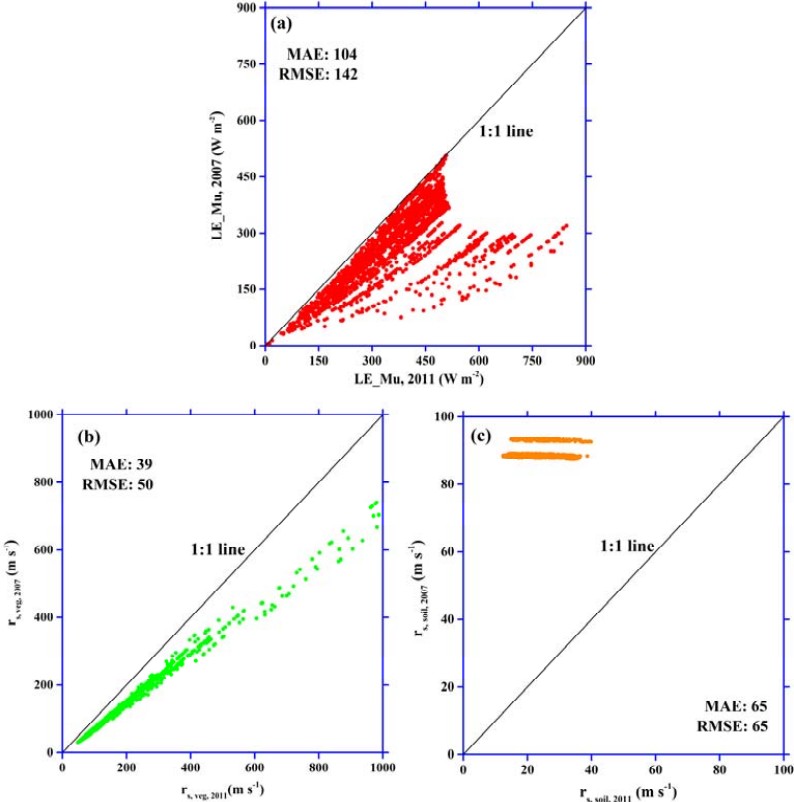

**Figure 9: Differences in latent heat flux (LE) estimates from modifications to the $r_s$ calculation when using PM_Mu algorithms: (a) LE, (b) parameterizations of the surface resistance of vegetation ($r_{s, veg}$) according to the old algorithms in Mu et al. (2007) and the improved algorithms in Mu et al. (2011), and (c) parameterizations of the soil resistances ($r_{s, soil}$) according to the old algorithms in Mu et al. (2007) and the improved algorithms in Mu et al. (2011).**
**Note: MAE and RMSE represent the mean absolute error and root mean squared error, respectively.**




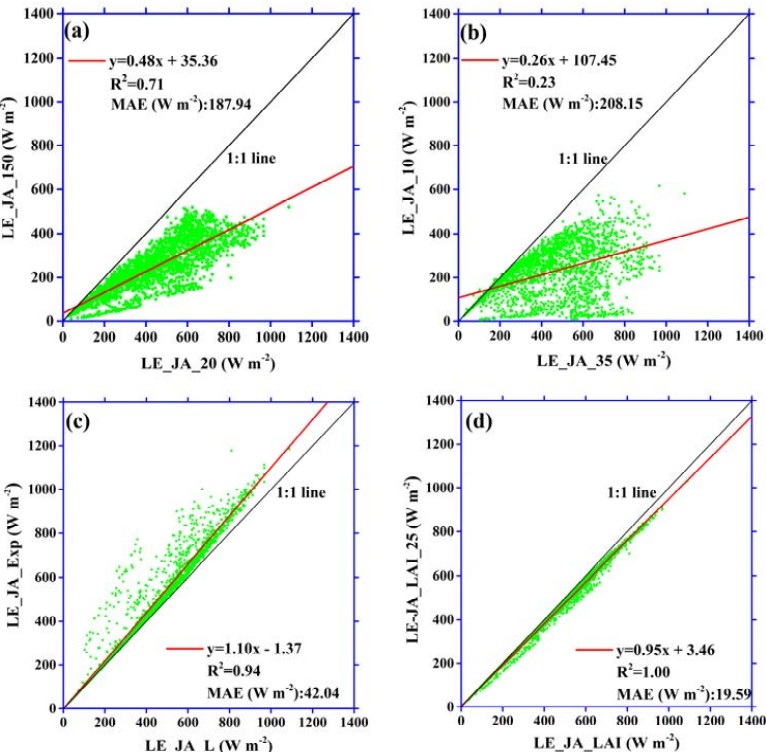

**Figure 10: Differences in latent heat flux (LE) estimates from modifications to the $r_s$ calculation in PM_JA: (a) modifying the minimum stomatal resistance ($r_{smin}$), (b) modifying the optimal temperature when parameterizing $f$ ($T_a$), (c) modifying the parameterization of $f$ ($VPD$), and (d) an LAI change of −25%.**

5   **Note: The LE values were estimated using PM_JA; MAE represents the mean absolute error; LE_JA_20 and LE_JA_150 represent the estimated LE values when setting rsmin to 20 and 150 s m−1, respectively; LE_JA_10 and LE_JA_35 represent the estimated LE values when parameterizing f (Ta) using optimal temperatures of 10 and 35 °C, respectively; LE_JA_L and LE_JA_Exp represent the estimated LE values when parameterizing f (VPD) using linear and exponential algorithms, respectively; and LE_JA_ LAI _25 was calculated by setting LAI to 0.75 times the LAI value of LE_JA_LAI.**



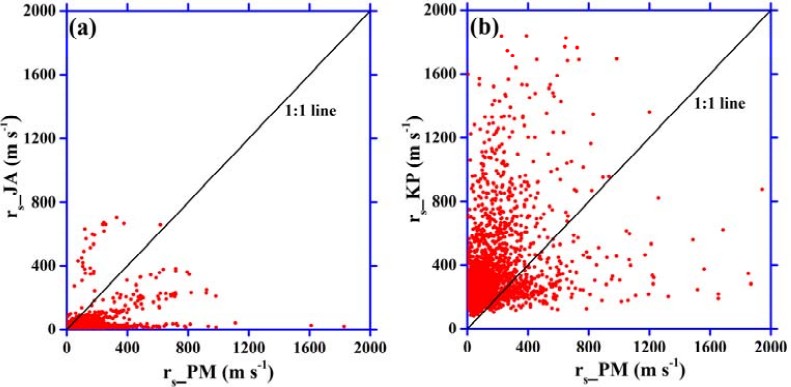

**Figure 11: Comparison of the estimated surface resistance, $r_s$, from the Jarvis (1976) ($r_s$_JA) and Katerji and Perrier (1983) ($r_s$_KP) methods and the inverse of the one-source PM equation ($r_s$_PM).**

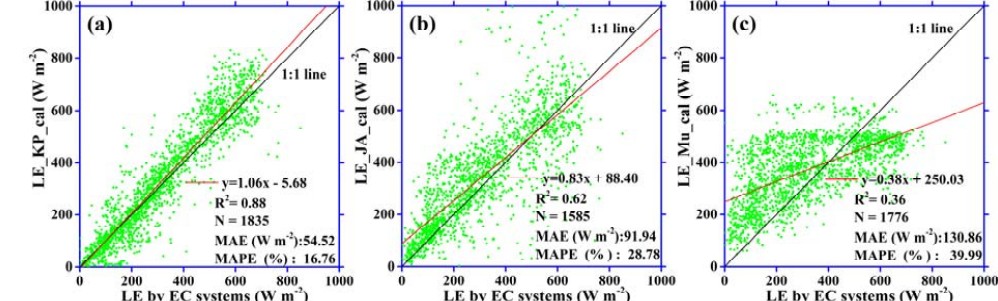

**Figure 12: Performance of the PM equation after optimizing the surface resistance parameterization with observational data. LE_JA_cal, LE_KP_cal, and LE_Mu_cal were estimated with resistances using the calibrated values in Table 1. MAE and MAPE represent the mean absolute error and absolute percent error, respectively. The temporal scale of the data is one half hour.**