# Peer review of "Uncertainty caused by resistances in evapotranspiration"

_Hydrology and Earth System Sciences, 2019_

## Referee Comment (RC1) · Anonymous Referee #1 · 17 Jun 2019

Zhao et al. compare simulations of evapotranspiration (ET) from a big-leaf (one-source) Penman-Monteith model with simulations from a two-source model and the three temperature (3T) model. Simulated ET is compared to measured fluxes from a number of eddy covariance sites in an oasis in Northwest China. Focus is given to uncertainties caused by the parameterization of resistances (surface and aerodynamic) in these formulations.

The study investigates a relevant scientific question, which is nicely introduced in the introduction section. However, I cannot recommend this manuscript to be published due to the following shortcomings:

- the comparison of the different approaches is reduced to a simple sensitivity analysis, resulting in very little scientific progress. The key message from Figures 8-12 is that

different models or a different parameterization of the same model give different results, which will be of little interest to the readers.

- Likewise, the discussion section does not provide a reasonable scientific contribution. For example, it is concluded (page 15, l.10f.) that the use of calibrated values would improve model performance. This statement is certainly true, but also obvious. I encourage the authors to repeat the analysis and focus on more relevant scientific questions, for example: - more complex/parameter-rich models are more likely to give accurate results, but are also more difficult to parameterize and to apply across sites. In that sense, is the use of more complex models justified in that case? Does better model performance outweigh the difficulty of finding the right parameter values? This is already discussed in the manuscript, but not clearly presented and not quantified. - what is a reasonable approach to estimate resistance values required in ET models? Which approach is most applicable here? Do the parameters have a mechanistic meaning, i.e. can the model be applied to other sites as well if some key biophysical properties are available? - are uncertainties mostly caused by surface resistance or aerodynamic resistance? What role do these individual resistances play under different environmental conditions?

- the presentation of the results is weak. Most of the figures show irrelevant or redundant information. For example, Figure 5 shows $R^2$ values of individual flux sites that will not be meaningful for most readers who do not know the sites. Figure 6 is supposed to show values of surface temperature but axes labels and units denote LE values in W m-2. Tables 4-6 show results of individual flux towers at a single point in time, which is of little interest to the readers.

specific comments:

Introduction

- page 2, l. 19: "Directly measuring surface or canopy resistance can also be difficult" is rather optimistic. I am not aware of any approaches capable of measuring canopy

resistance directly. - page 2, l. 20: there is nothing wrong with defining stomatal conductance as Cs, but I would strongly recommend to stick to the most common notation of gs, as it is simpler to read.

Methods

section 2.1.2: it needs become clearer why the formulation of Jarvis 1976 was chosen here instead of more recent stomatal conductance models such as Medlyn et al. 2011. I would not persist on taking the Medlyn model here, but it would be helpful for the reader to understand why the Jarvis model is taken, despite the fact that Medlyn is more often used in e.g. land surface models, and easier to parameterize (only 2-3 parameters)

- Eq. 3: change label to Eq. 3a,3b,3c,3d, which makes it easier to refer to

- Eq. 6: why are effects of atmospheric turbulence ("stability correction") not included here?

- p. 5, l. 17: since ra is calculated according to Eq. 6 in the Jarvis approach as well, I would show Eq. 6 directly after Eq. 3

- I wonder if rs in the Jarvis and KP parameterization is conceptually the same thing? rs in the Jarvis model is surface resistance only, whereas rs in the KP method includes both surface and aerodynamic resistances (Eq. 4). This could also contribute to the fact that rs as simulated by KP is much higher than the one given by Jarvis (Figure 8). In any case, differences between rs in Eq. 2 and 4 must be clarified, or a different notation used.

- throughout the manuscript, a clearer distinction between surface and aerodynamic resistance would be appropriate. In general, little attention is given to the aerodynamic part. I am sure many readers would be interested in the contribution of aerodynamic and surface resistances under different environmental conditions.

Results

[Figure]

- Figure 3: I do not think that the comparison of absolute LE fluxes is the best way of presenting the results. As indicated on page 9 l.7f., fluxes were Bowen ratio adjusted if energy balance closure is less than 80%. A critical discussion of the implications of this adjustment would be adequate.

- what is the justification for the sensitivity analysis as shown in Figure 10? In particular, what is the justification to assume an optimal temperature of 10degC for a C4 plant? Likewise, where do the different numbers of the rsmin come from? Why are previous estimates of rsmin so different (page 13 l. 25ff.)? These parameters will of course critically affect simulations of LE, but rather than just telling the reader what these different values would give in terms of LE, it would be be much more meaningful to discuss approaches to parameterize the models. E.g. how should we parameterize rsmin? This is a physiological parameter that can and should be measured, rather than optimized.

- Table 2: ordering the Table according to the site ID would make the Table easier to screen. Please also add the surface type as additional column. Please also remind the reader what year the column 'observation duration' is referring to.

- Figure 1: it would be clearer to show the extent of Fig. 1b in Fig. 1a as it is done for Fig. 1c and 1b.

---

## Referee Comment (RC2) · Anonymous Referee #2 · 9 Jul 2019

The paper titled 'Uncertainty caused by resistances in evapotranspiration' by Zhao et al aimed at quantifying the uncertainties surface resistance parameterization to understand and improving terrestrial evapotranspiration (ET) models. This is a much-needed idea, however, the presentation of the manuscript needs substantial improvement before being published in HESS. Very surprisingly, the authors did not attempt to review the literatures carefully. It seems they are either not well versed with the recent literature that emphasized on overcoming the resistance estimation uncertainties. The empirical resistance model of Jarvis and KP has no physical basis. Here are my suggestions and comments, which needs to be considered before being approved for publications. (1) Some recently published ET modeling and mapping studies that particularly addressed the challenges of resistance parameterizations (that deserves to be

considered here); for example,

Mallick et al. (2018). Bridging Thermal Infrared Sensing and Physically-Based Evapotranspiration Modeling: From Theoretical Implementation to Validation Across an Aridity Gradient in Australian Ecosystems, Water Resources Research, 54, 3409–3435. https://doi.org/10.1029/2017WR021357. Mallick et al. (2015). Reintroducing radiometric surface temperature into the Penman-Monteith formulation, Water Resources Research, 51, 6214–6243, http://doi.org/10.1002/2014WR016106. Garcia et al. (2013); Actual evapotranspiration in drylands derived from in-situ and satellite data: Assessing biophysical constraints. https://www.sciencedirect.com/science/article/abs/pii/S0034425712004828.

Morillas et al. (2013); Improving evapotranspiration estimates in Mediterranean drylands: The role of soil evaporation. https://agupubs.onlinelibrary.wiley.com/doi/pdf/10.1002/wrcr.20468.

Mallick et al. (2014). A surface temperature initiated closure (STIC) for surface energy balance fluxes, Remote Sensing of Environment, 141, 243 - 261. Bhattarai et al. (2019). An automated multi-model evapotranspiration mapping framework using remote sensing and reanalysis data. Remote Sensing of Environment, 229, 69 - 92. Gerhards et al. (2019). Challenges and Future Perspectives of Multi-/Hyperspectral Thermal Remote Sensing for Crop Water Stress Detection: A Review, Remote Sensing, 11(10), 1240; https://doi.org/10.3390/rs11101240. Bhattarai et al (2018). Regional evapotranspiration from image-based implementation of the Surface Temperature Initiated Closure (STIC1.2) model and its validation across an aridity gradient in the conterminous United States, Hydrology and Earth System Sciences, 22, 2311-2341, https://doi.org/10.5194/hess-22-2311-2018. Mallick, K., Trebs, I., Boegh, E., Giustarini, L., Schlerf, M., Drewry, D. T., et al. (2016). Canopy-scale biophysical controls of transpiration and evaporation in the Amazon Basin. Hydrology and Earth System Sciences, 20, 4237–4264. https://doi.org/10.5194/hess-20-4237-2016. Katerji et al. (2011), Parameterizing canopy resistance using mechanistic and semi‐empirical estimates of

hourly evapotranspiration: critical evaluation for irrigated crops in the Mediterranean, https://onlinelibrary.wiley.com/doi/abs/10.1002/hyp.7829

(2) Influence of the resistance parameterization on ET: Residual error analysis of ET with respect to resistance, soil moisture, VPD and net available energy (RN – G) needs to be discussed in detail. (3) How the resistance models performed under different soil moisture, VPD and radiation conditions? Without a detailed analysis, it would be difficult to assess the scientific value of the paper. (4) How the 3T model performed under different soil moisture, VPD and radiation conditions? (5) A Table of symbols and their unit for different models would greatly improve the readability of the manuscript. (6) Analysis of Sensible heat fluxes should also be included in a condensed manner.

(7) How 3T model avoids the parameterization of the resistances? This is a good side of the model. However, it needs to be described in a condensed manner.

I believe this manuscript can (and should) be improved substantially to give it good scientific quality.

---

## Author Comment (AC1) · 10 Jul 2019

Dear Reviewer:

Thank you for carefully reading the manuscript and providing constructive suggestions and comments. We address all of the comments point-by-point below. All of the revisions are highlighted in the new version of the manuscript.

NOTE: This revision of the manuscript (attached as supplement file) is based on comments from Reviewer 1 (because comments from Reviewer 2 were received on July 7, 2019); the newly added content is shown in blue, and any revised content is shown in red. Our answers to every question/comment are provided below.

Zhao et al. compare simulations of evapotranspiration (ET) from a big-leaf (one-source)

Penman-Monteith model with simulations from a two-source model and the three temperature (3T) model. Simulated ET is compared to measured fluxes from a number of eddy covariance sites in an oasis in Northwest China. Focus is given to uncertainties caused by the parameterization of resistances (surface and aerodynamic) in these formulations. The study investigates a relevant scientific question, which is nicely introduced in the introduction section. However, shortcomings existed in the manuscript and I cannot recommend publication.

Responses:

Thank you for your positive evaluation, especially the comment on the introduction section. We attempted to revise the manuscript with the help of your comments and suggestions, which hopefully has improved the quality of the manuscript.

Comment 1: The comparison of the different approaches is reduced to a simple sensitivity analysis, resulting in very little scientific progress. The key message from Figures 8-12 is that different models or a different parameterization of the same model give different results, which will be of little interest to the readers.

Responses:

Thank you.

As described in the introduction section, the topic of simulation uncertainty has high academic and practical relevance. In fact, our manuscript has received attention from the research community with more than > 390 views in last 50 days (please see the statistics at https://www.hydrol-earth-syst-sci-discuss.net/hess-2019-160/#discussion).

The main goal of our manuscript is to identify the sources of uncertainty in resistance parameterizations in ET estimates. We quantified these uncertainties by comparing resistance parameterizations with different complexities with the one- and two-source Penman-Monteith (PM) equations. As such, we can investigate how the model structure and the parameterization process will affect the ET estimates. The method used

in our study is simple, but the results are intuitive and straightforward. Scientific research does not necessarily require complicated methods. In other words, the figures mentioned above were quantitative explanations that show how the model structure and the parameterization process will affect the ET estimates and thus, improve our understanding of the uncertainty caused by resistance when estimating ET. Our reply to specific comment 10 shows an example that due to difficulty in observing physiological parameters, these values may exhibit substantial difference and cause uncertainty in ET estimation.

Nonetheless, in this revision, we reorganized the results and discussion sections based on your comments, and Figures 8-12 were revised accordingly. For example, the original Figs. 8 and 11 were merged to avoid redundant information. Please see the revised manuscript for details.

Comment 2: Likewise, the discussion section does not provide a reasonable scientific contribution. For example, it is concluded (page 15, l.10f.) that the use of calibrated values would improve model performance. This statement is certainly true, but also obvious. I encourage the authors to repeat the analysis and focus on more relevant scientific questions, for example: - more complex/parameter-rich models are more likely to give accurate results, but are also more difficult to parameterize and to apply across sites. In that sense, is the use of more complex models justified in that case? Does better model performance outweigh the difficulty of finding the right parameter values? This is already discussed in the manuscript, but not clearly presented and not quantified. - what is a reasonable approach to estimate resistance values required in ET models? Which approach is most applicable here? Do the parameters have a mechanistic meaning, i.e. can the model be applied to other sites as well if some key biophysical properties are available? - are uncertainties mostly caused by surface resistance or aerodynamic resistance? What role do these individual resistances play under different environmental conditions?

Responses:

Thank you for this detailed comment.

As you mentioned, because the section already discussed problems related to the impact of models with different complexities on ET estimation, we reorganized the discussion section and focused more on these scientific questions. However, several questions are currently beyond our analysis. For example, we analyzed the impact of a change in wind speed on the aerodynamic resistance, but aerodynamic resistance is difficult to determine, and the uncertainty from aerodynamic resistance is not a major topic in this study. Hence, we could not identify which factor (surface resistance or aerodynamic resistance) contributes more to the uncertainty in ET estimation.

In this revision of the manuscript, we revised the discussion section. In the beginning (Section 5.1 Uncertainty caused by resistance in ET estimates), we discuss the general influence of different model complexities and parameterization processes on the ET estimates and clearly presented the results. Section 5.2 still discusses the uncertainty in canopy resistance estimation but was also revised, because most of the discussed biophysical variables are commonly used in canopy resistance estimation, and the problems discussed in this section are faced by the research community. Similarly, Section 5.3 (Possible solution for reducing uncertainty in resistance parameterization) was kept and revised. We believe that the discussed biophysical boundaries, which were deduced from site experiments (observations) rather than from an understanding of their mechanistic theory, can cause substantial uncertainty in ET estimation. Therefore, calibrations of such biophysical boundaries are necessary. However, the calibrations are performed by trial and error and require additional observation datasets, which may be impossible to obtain in most applications. With these discussions, we hope to provide insightful information for interested readers. Please see the revised manuscript for details.

Comment 3: The presentation of the results is weak. Most of the figures show irrelevant or redundant information. For example, Figure 5 shows R̈Ȩ2 values of individual flux sites that will not be meaningful for most readers who do not know the sites. Figure 6

is supposed to show values of surface temperature but axes labels and units denote LE values in W m-2. Tables 4-6 show results of individual flux towers at a single point in time, which is of little interest to the readers.

Responses:

Thank you for the detailed comment.

Figure 5 was intended (figure 6 in this revision) to evaluate models with different complexities and parameterizations at different maize sites because the study area was a maize-dominated heterogeneous oasis, and the maize fields exhibited varied plant biophysics and soil moisture content conditions. We redrew the figure by showing the model performance (MAPE) in the LAI-soil moisture space, which may provide intuitive information to readers even they do not know the sites, as follows:

[insert figure 6 here]

Figure 6 (figure 5 in this revision) was used to show the differences in LE estimates when using different temperature parameterizations. To prevent misunderstandings, we revised the notes in the figure, i.e., changing "Ts (Fluke)" to "LE_3T: LST (LE_3T: Fluke)" , as follows:

[insert figure 5 here]

Tables 3-6 were used as examples in several places in the manuscript. We believe they are necessary, but we moved them to a supplemental file.

In this revision, we reorganized and revised (enhanced) the results section. First, we present the ET estimates from models with different complexities and parameterizations (Section 4.1 Characteristics of the LE estimates). We then show the differences between the estimates (Section 4.2 Differences among the LE estimates and model performance), and we finally presented the sources of the differences between the estimates (4.3 Sources of difference among the LE estimates and model performance). Please see the revised manuscript for details.

Specific comments:

Comment 1: Introduction. page 2, l. 19: "Directly measuring surface or canopy resistance can also be difficult" is rather optimistic. I am not aware of any approaches capable of measuring canopy.

Responses:

Thank you. Based on this comment, we revised this sentence as follows:

Surface or canopy resistance cannot be measured directly, and these terms are often estimated and scaled from the leaf stomatal resistance or its inverse, the leaf stomatal conductance (gs), using the leaf area index (LAI) (Wang and Dickinson, 2012; Schymanski and Or, 2017) or estimated from flux-tower-based meteorological observations and the inversed one-source PM equation.

Comment 2: Introduction. page 2, l. 20: there is nothing wrong with defining stomatal conductance as Cs, but I would strongly recommend to stick to the most common notation of gs, as it is simpler to read.

Responses:

Thank you. Based on this comment, we changed Cs to gs (an example is shown in our reply to specific comment 1). Please see the revised manuscript for details.

Comment 3: Methods. section 2.1.2: it needs become clearer why the formulation of Jarvis 1976 was chosen here instead of more recent stomatal conductance models such as Medlyn et al. 2011. I would not persist on taking the Medlyn model here, but it would be helpful for the reader to understand why the Jarvis model is taken, despite the fact that Medlyn is more often used in e.g. land surface models, and easier to parameterize (only 2-3 parameters).

Responses:

Thank you. Based on this comment, we added several sentences to Section 2.1 to

explain the reason for using the formulation of Jarvis (1976) as follows:

For the one-source PM equation (Eq. 1), rs was calculated using two classical methods: the method of Jarvis (JA) (1976) and the method of Katerji and Perrier (KP) (1983) (hereby abbreviated as PM_JA and PM_KP, respectively). The empirical KP method requires fewer inputs than the JA method and can be used easily in practical applications, but the variables in the JA method have clear physical meanings that may better represent actual conditions. Therefore, we can address questions such as if more accurate results are likely to be obtained by more complex models. If the answer is yes, is a more complex model more difficult to parameterize and apply? Furthermore, algorithms with substantial differences are used to parameterize a given variable in the JA method; thus, we can investigate the uncertainty generated by the nonunique parameterizations. In addition, to accurately describe heat or water vapor transfer, the PM equation can be extended to include two or more sources depending on the configuration of the resistance networks. Therefore, a modified two-source PM equation that was proposed for RS applications (Mu et al. 2011) (abbreviated as PM_Mu) was also used to estimate ET as well as discuss the problems described above (see Section 2.2).

Comment 4: Methods. Eq. 3: change label to Eq. 3a, 3b, 3c, 3d, which makes it easier to refer to.

Responses:

Thank you. Based on this comment, we labeled the equations separately as suggested. Please see the revised manuscript for details.

Comment 5: Methods. Eq. 6: why are effects of atmospheric turbulence ("stability correction") not included here?

Responses:

Thank you. In this study, we mainly focused on the uncertainty caused by rs. Because

the calculation of ra depends on several parameters that are difficult to obtain accurately (e.g., atmospheric stratification), ra was estimated without considering the effect of atmospheric turbulence to reduce the uncertainty in ra estimation. Based on this comment, we added such information to Section 2.1 as follows:

2 Description of ET models

2.1 One-source PM equation and its parameterization

The PM equation is based on a single big-leaf assumption (one-source) and the energy budget closure, as follows (Monteith, 1965):

[insert Eq. (1) here]

where Rn is the net radiation; . . .; rs is the surface resistance and ra is the aerodynamic resistance.

In this study, we focus on the uncertainty caused by rs rather than ra. To reduce the uncertainty in ra estimation (e.g., lack of detailed atmospheric stratification data), ra was calculated without considering the effect of atmospheric turbulence in the one-source PM equation using Equation (2) (Brutsaert and Stricker, 1979; Irmak et al., 2008).

[insert Eq. (2) here]

Comment 6: Methods. p. 5, l. 17: since ra is calculated according to Eq. 6 in the Jarvis approach as well, I would show Eq. 6 directly after Eq. 3.

Responses:

Thank you. Based on this comment, we adjusted the location of Eq. 6 and placed directly after Eq. 1 when introducing the PM equation (please see our reply to comment 5 for details). The orders of the related equations was also revised (please see the revised manuscript for details).

Comment 7: Methods. I wonder if rs in the Jarvis and KP parameterization is conceptually the same thing? rs in the Jarvis model is surface resistance only, whereas rs in the KP method includes both surface and aerodynamic resistances (Eq. 4). This could also contribute to the fact that rs as simulated by KP is much higher than the one given by Jarvis (Figure 8). In any case, differences between rs in Eq. 2 and 4 must be clarified, or a different notation used.

Responses:

Thank you. The parameterization method for rs differs significantly different between the Jarvis and KP methods. In the KP method, it uses the aerodynamic resistance as an input. Nonetheless, we believe that the purpose of the two methods is to calculate rs for the one-source PM equation.

Comment 8: Methods. Throughout the manuscript, a clearer distinction between surface and aerodynamic resistance would be appropriate. In general, little attention is given to the aerodynamic part. I am sure many readers would be interested in the contribution of aerodynamic and surface resistances under different environmental conditions.

Responses:

Thank you. This problem requires further study. However, we neglected the uncertainty from aerodynamic resistance in this study because is difficult to determine and its calculation depends on several parameters that are difficult to obtain accurately, such as the roughness height and atmospheric stratification.

Comment 9: Results. Figure 3: I do not think that the comparison of absolute LE fluxes is the best way of presenting the results. As indicated on page 9 l.7f., fluxes were Bowen ratio adjusted if energy balance closure is less than 80%. A critical discussion of the implications of this adjustment would be adequate.

Responses:

[Figure]

Thank you. In this study, the observed LE fluxes are corrected using the Bowen ratio closure method if the closure rate (represented by (LE+H)/(Rn-G)) is less than 0.8. Under such conditions, some of the LE estimates are not completely independent from the observations corrected using the Bowen ratio closure method, as raised in the comment. The comparison method is commonly adopted to validate modeled ETs (e.g., Ershadi et al., 2014, Agricultural and Forest Meteorology, 187, 46–61; Jiang and Ryu, 2016, Remote Sensing of Environment, 186, 528–547), including those retrieved from methods that contain the "Rn-G" term, such as the Penman-Monteith equation and the Priestley-Taylor method.

Based on this comment, we added text to discuss the problems in the last paragraph in Section 4.2 as follows:

As shown in Figs. 2 and 7, the investigation data covered different phenological stages and weather conditions during the 2012 growing season. For example, the daytime (7:00-19:00 GMT+8) solar radiation varied from 0 to 1055 W m$-2$, with a mean value of 464 W m$-2$ and an SD of 307 W m$-2$ (Fig. 7a). The mean wind speed was 1.5 m s$-1$ with maximum, minimum, and SD values of 5.8, 0.2, and 1.1 m s$-1$, respectively (Fig. 7b). The average temperature was 23.1 °C, with maximum, minimum, and SD values of 33.1, 8.4, and 4.6 °C, respectively (Fig. 7c). Although approximately half of the EC-observed LE values were adjusted with the Bowen ratio to achieve energy balance closure, which may make the ET estimates incompletely independent from the corrected observations, because the "Rn-G" term was used in both the adjustment and the model estimation, the validation results at a half-hourly scale, as shown in the previous sections, were recorded at different phenological stages and under various atmospheric conditions during the growing season, indicating a meaningful comparison.

Comment 10: Results. What is the justification for the sensitivity analysis as shown in Figure 10? In particular, what is the justification to assume an optimal temperature of 10degC for a C4 plant? Likewise, where do the different numbers of the rsmin come

from? Why are previous estimates of rsmin so different (page 13 l. 25ff.)? These parameters will of course critically affect simulations of LE, but rather than just telling the reader what these different values would give in terms of LE, it would be much more meaningful to discuss approaches to parameterize the models. E.g. how should we parameterize rsmin? This is a physiological parameter that can and should be measured, rather than optimized.

Responses:

Thank you. The value of minimum resistance for a leaf (rsmin) can be measured. However, measured values for plant species under different water and climatic combinations are rare. In this study, such data were unavailable; therefore, we cited these physiological values published values in a maize field that was adjacent to our study area.

In our calculation, the rsmin was set to 20 s m−1 according to Li et al. (Journal of Hydrology, 2013, 489, 124–134) (also Mu et al., Remote Sensing of Environment, 2011, 115, 1781–1800). The different value, i.e., rsmin=150 s m−1, was used to perform the sensitive analysis, and this value was cited in another study from the same research group (Li et al., Agricultural Water Management, 2016, 178, 314–324). Similarly, the optimal temperature values used in this study (i.e., 35 and 10 °C) were applied in the arid Heihe River Basin (or adjacent areas). We used 35 °C for the optimal temperature according to Hu and Jia (Remote Sensing, 2015, 7, 3056−3087), but 10 °C was adopted for a sensitivity comparison. The value of 10 °C for maize was optimized based on field observations by Li et al. (Journal of Hydrology, 2013, 489, 124–134).

I communicated and discussed with Dr. Li personally. The reason he used an optimal value, rather than the observed one, is that although the observed values covering 4 years, the observation could not perform every day and year around (not consecutive), but the rsmin (or the optimal temperature) value varied in different phenological stages. In addition, one could not guarantee the measured physiological parameter, e.g., the

rsmin, is the minimum value.

Considering difficulties in field measurement and substantial difference exited in the same physiological parameter optimized from difference observational data set, we believe the sensitivity analysis in our study was necessary and may provide readers with useful information.

Comment 11: Results. Table 2: ordering the Table according to the site ID would make the Table easier to screen. Please also add the surface type as additional column. Please also remind the reader what year the column 'observation duration' is referring to.

Responses:

Thank you. We revised Table 2 as follows:

[insert Table 2 here]

Comment 12: Results. Figure 1: it would be clearer to show the extent of Fig. 1b in Fig. 1a as it is done for Fig. 1c and 1b.

Responses:

Thank you. We revised the figure as follows:

[insert figure 1 here]

———————————————

[Figure]

Figure 1: Study area: (a) locations of the Zhangye Oasis and the Heihe River Basin in China, (b) 21 EC flux towers in HiWATER, and (c) detailed locations of the 17 EC systems within the key experimental area in the Zhangye Oasis. (b) and (c) were revised from Ma et al. (2015); AWS and LAS are abbreviations for automatic weather station and large-aperture scintillometer, respectively.

[Figure]

**Fig. 1.**

Figure 5: Comparison of the half-hourly ET estimates from the 3T model and measurements at a residential flux site. MAE and MAPE represent the mean absolute error and absolute percent error, respectively. The red crosses represent the estimated LE from the 3T model (LE_3T: LST) using the reference temperatures from the observed land surface temperatures (LSTs) at the 19th EC system, whereas the green dots are the LEs estimated by the 3T model (LE_3T: Fluke) using Fluke thermal-image-based temperatures as the reference temperatures.

**Fig. 2.**

[Figure]

Figure 6: Performance of different models at different maize sites with different leaf area index (LAI) and soil moisture conditions in terms of mean absolute percent error (MAPE): (a) PM_JA method, (b) PM_KP method, (c) PM_Mu method, and (d) the 3T model. Note: the number represents the EC system ID in Figure 1c; the data are from Figure 4; panels a, b, and c have the same color legend.

Fig. 3.

**Table 2. Details of the 17 eddy covariance (EC) systems in the key experimental area shown in Fig. 1c.**

| EC ID | Sensor type & manufactures | Sensor height (m) | Observation duration in 2012 | Surface condition | LAI ($m^2 m^{-2}$) | Soil moisture (%) |
|---|---|---|---|---|---|---|
| 1 | Gill/Li7500A, Gill, UK/Li-cor, USA | 3.8 | Jun. 16 to Sep. 17 | vegetable field | 1.83 | 36.04 |
| 2 | CSAT3/Li7500, Campbell/Li-cor, USA | 3.7 | Jun. 15 to Sep. 21 | maize | 2.49 | 20.73 |
| 3 | Gill/Li7500A, Gill, UK/Li-cor, USA | 3.8 | Jun. 25 to Sep. 18 | maize | 2.35 | 36.57 |
| 4 | | 4.2 (6.2 after Aug.19) | May 31 to Sep. 17 | residential area | – | 18.65 |
| 5 | CSAT3/Li7500A, Campbell/Li-cor, USA | 3 | Jun. 3 to Sep. 18 | maize | 2.50 | 29.83 |
| 6 | | 4.6 | May 28 to Sep. 21 | maize | 2.39 | 31.57 |
| 7 | | 3.8 | May 29 to Sep. 18 | maize | 2.41 | 29.59 |
| 8 | | 3.2 | May 28 to Sep. 21 | maize | 2.58 | 31.53 |
| 9 | Gill/Li7500A, Gill, UK/Li-cor, USA | 3.9 | Jun. 25 to Sep. 17 | maize | 3.42 | 35.99 |
| 10 | | 4.8 | Jul. 5 to Sep. 17 | maize | 2.62 | 30.14 |
| 11 | | 3.5 | May 29 to Sep. 18 | maize | 2.26 | 27.12 |
| 12 | CSAT3/Li7500, Campbell/Li-cor, USA | 3.5 | May 28 to Sep. 21 | maize | 2.44 | 20.82 |
| 13 | | 5 | May 27 to Sep. 20 | maize | 2.27 | 22.62 |
| 14 | | 4.6 | May 30 to Sep. 17 | maize | 2.53 | 21.83 |
| 15 | | 4.5 | May 25 to Dec. 30 | maize | 3.14 | 28.45 |
| 16 | Gill/Li7500, Gill, UK/Li-cor, USA | 4.9 | Jul. 2 to Sep. 17 | maize | 2.61 | 28.76 |
| 17 | CSAT3/EC150, Campbell, USA | 7 | May 31 to Sep. 17 | orchard | 1.65 | 28.90 |

Note: 1. All of the sensor types are open-path, and related information was cited from Liu et al. (2016);

2. The sampling frequency of the EC systems was 10 Hz, and the EC data were post processed, quality controlled, recorded every 30 min on average by Liu et al. (2016) and distributed by the HiWATER project;

3. LAI and soil moisture values were averaged using corresponding data provided by the HiWATER project.

**Fig. 4.**

---

## Author Comment (AC2) · 23 Jul 2019

**Reply to Reviewer 2**

**Manuscript title**: Uncertainty caused by resistances in evapotranspiration

Dear Reviewer:

Thank you for carefully reading our manuscript and providing constructive suggestions and comments.

Below, we address all comments carefully point-by-point. All revisions are highlighted, i.e., newly added content is shown in blue, and any revised content is shown in red.

Our answers to every question/comment are provided below.

**Reviewer: 2**

The paper titled 'Uncertainty caused by resistances in evapotranspiration' by Zhao et al aimed at quantifying the uncertainties surface resistance parameterization to understand and improving terrestrial evapotranspiration (ET) models. This is a much-needed idea, however, the presentation of the manuscript needs substantial improvement before being published in HESS. Very surprisingly, the authors did not attempt to review the literatures carefully. It seems they are either not well versed with the recent literature that emphasized on overcoming the resistance estimation uncertainties. The empirical resistance model of Jarvis and KP has no physical basis. Here are my suggestions and comments, which needs to be considered before being approved for publications. I believe this manuscript can (and should) be improved substantially to give it good scientific quality.

**Responses:**

Thank you for appreciating our work and considering that the studied topic is a much-needed idea. We also thank you for pointing out the potential literature that we not included. Acknowledging this fact, we attempted to revise the manuscript with the help of your constructive comments and suggestions. With detailed replies to your comments, we believe that our manuscript has been improved substantially.

**Comment 1**:

Some recently published ET modeling and mapping studies that particularly addressed the challenges of resistance parameterizations (that deserves to be considered here); for example,

Mallick et al. (2018). Bridging Thermal Infrared Sensing and Physically Based Evapotranspiration Modeling: From Theoretical Implementation to Validation Across an Aridity Gradient in Australian Ecosystems, Water Resources Research, 54, 3409–3435. https://doi.org/10.1029/2017WR021357.

Mallick et al. (2015). Reintroducing radiometric surface temperature into the Penman-Monteith formulation, Water Resources Research, 51, 6214–6243, http://doi.org/10.1002/2014WR016106.

Garcia et al. (2013); Actual evapotranspiration in drylands derived from in-situ and satellite data: Assessing biophysical constraints. https://www.sciencedirect.com/science/article/abs/pii/S0034425712004828.

Morillas et al. (2013); Improving evapotranspiration estimates in Mediterranean drylands: The role of soil evaporation. https://agupubs.onlinelibrary.wiley.com/doi/pdf/10.1002/wrcr.20468.

Mallick et al. (2014). A surface temperature initiated closure (STIC) for surface energy balance fluxes, Remote Sensing of Environment, 141, 243 - 261.

Bhattarai et al. (2019). An automated multi-model evapotranspiration mapping framework using remote sensing and reanalysis data. Remote Sensing of Environment, 229, 69 - 92.

Gerhards et al. (2019). Challenges and Future Perspectives of Multi-/Hyperspectral Thermal Remote Sensing for Crop Water Stress Detection: A Review, Remote Sensing, 11(10), 1240; https://doi.org/10.3390/rs11101240.

Bhattarai et al (2018). Regional evapotranspiration from image-based implementation of the Surface Temperature Initiated Closure (STIC1.2) model and its validation across an aridity gradient in the conterminous United States, Hydrology and Earth System Sciences, 22, 2311-2341, https://doi.org/10.5194/hess-22-2311-2018.

Mallick, K., Trebs, I., Boegh, E., Giustarini, L., Schlerf, M., Drewry, D. T., et al. (2016). Canopy-scale biophysical controls of transpiration and evaporation in the Amazon Basin. Hydrology and Earth System Sciences, 20, 4237–4264. https://doi.org/10.5194/hess-20-4237-2016.

Katerji et al. (2011), Parameterizing canopy resistance using mechanistic and semiâA˘ Rempirical estimates of hourly evapotranspiration: critical evaluation for irrigated crops in the Mediterranean, https://onlinelibrary.wiley.com/doi/abs/10.1002/hyp.7829.

**Responses:**

Thank you for referring to these detailed publications.

In accordance with this comment, we added these publications within the appropriate sections (as well as citations in reference section) in this revised version. Some examples are as follows:

Although $r_a$ can be difficult to determine because its calculation depends on certain parameters that are difficult to accurately obtain, such as the roughness height, zero-plane displacement, and atmospheric stratification (Brutsaert and Stricker, 1979; Gerhards et al., 2019), the uncertainty from $r_a$ is often neglected.

Therefore, at low LAI (e.g., < 2), relatively high uncertainties were expected from single source models, such as those typically used in the PM equation (e.g., Farahani and Bausch, 1995; Lafleur and Rouse, 1990; Morillas et al., 2013).

However, surface or canopy resistances embody complex processes and are difficult to estimate accurately from RS data because they are controlled by numerous factors, such as wind speed (Su, 2002; Sánchez et al., 2008), vegetation type, biophysics, canopy architecture, soil texture and soil water availability (Leuning, 1995; Shuttleworth and Gurney, 1990; Katerji et al., 2011; García et al., 2013; Lehmann et al., 2018).

To avoid the issue of parameterizing resistances, several methods have been proposed to estimate ET without such parameterizations, such as the three-temperature (3T) model (Qiu et al., 2006; Xiong et al., 2015; Wang Y. et al., 2016), the surface temperature initiated closure model (Mallick et al., 2014, 2015, 2016, 2018; Bhattarai et al., 2018), the Priestley-Taylor method (Priestley and Taylor, 1972), the triangle or trapezoidal method (Price, 1990; Long and Singh, 2012), the complementary relationship model (Ma et al., 2019), and the surface renewal method (Paw U et al., 1995).

New references were also added (please note that Katerji et al. (2011) was in our previous manuscript):

Bhattarai, N., Mallick, K., Brunsell, N. A., Sun, G., and Jain, M.: Regional evapotranspiration from an

image-based implementation of the Surface Temperature Initiated Closure (STIC1. 2) model and its validation across an aridity gradient in the conterminous US, Hydrol. Earth Syst. Sci., 22, 2311–2341, https://doi.org/10.5194/hess-22-2311-2018, 2018.

Bhattarai, N., Mallick, K., Stuart, J., Vishwakarma, B. D., Niraula, R., Sen, S., and Jain, M.: An automated multi-model evapotranspiration mapping framework using remotely sensed and reanalysis data, Remote Sens. Environ., 229, 69–92, https://doi.org/10.1016/j.rse.2019.04.026, 2019.

García, M., Sandholt, I., Ceccato, P., Ridler, M., Mougin, E., Kergoat, L., Morillas, L., Timouk, F., Fensholt, R., and Domingo, F.: Actual evapotranspiration in drylands derived from in-situ and satellite data: Assessing biophysical constraints, Remote Sens. Environ., 131, 103–118, https://doi.org/10.1016/j.rse.2012.12.016, 2013.

Gerhards, M., Schlerf, M., Mallick, K., and Udelhoven, T.: Challenges and Future Perspectives of Multi-/Hyperspectral Thermal Infrared Remote Sensing for Crop Water-Stress Detection: A Review, Remote Sens., 11, 1240, https://doi.org/10.3390/rs11101240, 2019.

Mallick, K., Jarvis, A. J., Boegh, E., Fisher, J. B., Drewry, D. T., Tu, K. P., Hook, S. J., Hulley, G., Ardö, J., and Beringer, J.: A Surface Temperature Initiated Closure (STIC) for surface energy balance fluxes, Remote Sens. Environ., 141, 243–261, https://doi.org/10.1016/j.rse.2013.10.022, 2014.

Mallick, K., Boegh, E., Trebs, I., Alfieri, J. G., Kustas, W. P., Prueger, J. H., Niyogi, D., Das, N., Drewry, D. T., and Hoffmann, L.: Reintroducing radiometric surface temperature into the Penman-Monteith formulation, Water Resour. Res., 51, 6214–6243, https://doi.org/10.1002/2014wr016106, 2015.

Mallick, K., Trebs, I., Boegh, E., Giustarini, L., Schlerf, M., Drewry, D. T., Hoffmann, L., RANDOW, C. v., Kruijt, B., and Araùjo, A.: Canopy-scale biophysical controls of transpiration and evaporation in the Amazon Basin, Hydrol. Earth Syst. Sci., 20, 4237–4264, https://doi.org/10.5194/hess-20-4237-2016, 2016.

Mallick, K., Toivonen, E., Trebs, I., Boegh, E., Cleverly, J., Eamus, D., Koivusalo, H., Drewry, D., Arndt, S. K., and Griebel, A.: Bridging Thermal Infrared Sensing and Physically-Based Evapotranspiration Modeling: From Theoretical Implementation to Validation Across an Aridity Gradient in Australian Ecosystems, Water Resour. Res., 54, 3409–3435, https://doi.org/10.1029/2017wr021357, 2018.

Morillas, L., Leuning, R., Villagarcía, L., García, M., Serrano-Ortiz, P., and Domingo, F.: Improving evapotranspiration estimates in Mediterranean drylands: The role of soil evaporation, Water Resour. Res., 49, 6572–6586, https://doi.org/10.1002/wrcr.20468, 2013.

**Comment 2**:

Influence of the resistance parameterization on ET: Residual error analysis of ET with respect to resistance, soil moisture, VPD and net available energy (RN-G) needs to be discussed in detail.

**Responses:**

Thank you for the valuable comment.

In fact, we discussed the influence of the resistance parameterization on ET in Section 5.1 in our previous manuscript, although not in terms of residual error analysis used by Mallick et al. (Water Resources Research, 2018, 54, 3409–3435). Specifically, we discussed the influence of the resistance parameterizations under different physiological threshold value scenarios via the Jarvis model. We believe that, although the physical basis of the Jarvis model is not as perfect as that of other methods (such as

Medlyn et al. (2011) mentioned by reviewer 1), the inputs require certain types of physiological variables. Because measurements of such physiological variables are rare, we cited two values for a given physiological variable to perform the analysis, e.g., changing the minimum resistance, $r_{smin}$, from 20 to 150 s m$^{-1}$ to investigate its impact on ET. In addition, these threshold values were carefully selected from the literature. Under such conditions, figure 10 could explain such a change in ET estimates.

In accordance with this comment, we analyzed the effects of $r_s$ on ET under the one- and two-source PM equations in terms of the residual ET error. Figures 8 and 9 were added to the revision in combination with the corresponding context:

4.3 Sources of the differences among the LE estimates and model performance

Notably, the different LE estimates and model performance of the one-source PM equations were caused by the different resistance parameterizations (i.e., the difference between Eqs. (3) and (5) in Section 2.1). Figure 8a shows the large differences between $r_s$_JA, which was estimated in accordance with the methods of Jarvis (1976), and $r_s$_KP, which was estimated in accordance with the methods of Katerji and Perrier (1983). Specifically, if the $r_s$ values greater than 1000 s m$^{-1}$ in figure 8a are assumed to be outliers, the mean value of $r_s$_JA was 33 s m$^{-1}$, with maximum and minimum values of 705 and 9 s m$^{-1}$, respectively, whereas the mean value of $r_s$_KP was 300 s m$^{-1}$, with maximum and minimum values of 998 and 80 s m$^{-1}$, respectively. The average difference between the $r_s$_JA and $r_s$_KP estimates was 267 s m$^{-1}$, which led to a difference of 194 W m$^{-2}$ in the LE estimates. Residual error analysis of LE (observation minus estimation) with respect to the $r_s$ in figures 8b and 8c revealed that $r_s$_KP was generally underestimated, whereas $r_s$_JA was mostly overestimated (especially when $r_s$_JA < 200 s m$^{-1}$). Furthermore, quantification and parameterization of the surface resistance are difficult. For example, if the JA resistance method is applied at the residential site (No. 4 at 12:30 GMT+8 on July 10), the value of $r_s$ is 288 s m$^{-1}$, leading to an LE value of 282 W m$^{-2}$, which was 107 W m$^{-2}$ greater than the EC observation. Thus, the resistance estimation method and its parameterization can be major sources of uncertainty in ET estimates.

…

5 Discussion

5.1 Uncertainty caused by resistance in ET estimates

It is generally recognized that a complex model that can physically represent additional details of a system is more likely to give accurate estimations than a less complex model is. The results from the one-source PM equations support this idea. Because the results presented in Section 4 showed that the LE_JA values, which were estimated using the surface resistance parameterized from the physically based JA method, were more similar in terms of the MAPE to the observations than were the LE_KP values, which were estimated using the surface resistance parameterized from the empirical KP method. Nonetheless, compared to the variation in residual LE error versus $r_s$_JA (Figs. 8b and 8c), a relatively small variation in residual LE error versus $r_s$_KP indicates a stronger correlation between the LE_KP and observed LE values ($R^2 = 0.9$ in Fig. 4b). In addition, the complete one-source PM_JA method requires more parameters, such as the soil water and LAI (Tables S1 and S2), than does the PM_KP method, so LE cannot be estimated if a necessary variable is missing for a given time. This idea explains why the number of LE_KP estimates from the PM_KP method was greater than that from the PM_JA method (Fig. 4).

However, the relatively complex two-source PM equation performed more poorly than did the one-source PM equation, as shown in Section 4.2, which is likely due to the two-source PM equation

requiring overcomplicated parameterizations. For example, the resistance parameterizations described by Mu et al. (2007) were improved in the report of Mu et al. (2011). In particular, the calculation of the surface resistance for vegetation involved different parameterizations of canopy conductance ($g_c$) (Table 3); the calculation of the surface resistance featured a $r_{totc}$ value of 107 s m$^{-1}$ (in Eq. 13); and the constant $\beta$ in the soil-evaporation estimation was changed from 100 to 200 in the improved algorithm (Eq. 15). These modifications created different resistance values and LE estimates (abbreviated as LE_Mu_2007 and LE_Mu_2011) (Figs. 9a to 9c); however, these parameter changes are only locally optimized and might not generalize well. The LE values after the modifications were, on average, 104 W m$^{-2}$ greater than those before the modification, and the root mean square error (RMSE) between LE_Mu_2007 and LE_Mu_2011 was 142 W m$^{-2}$ (Fig. 9a). This difference was caused by the improvement in the resistance parameterizations. As shown in figures. 9b and 9c, the modification of the canopy conductance caused a difference of 39 s m$^{-1}$ between the estimated canopy resistances, whereas the modification of $r_{totc}$ produced a difference of 65 s m$^{-1}$ in soil surface resistance. If the $\beta$ value remained at 100 in the improved algorithm, the estimated LE values would have had an average difference of only 2 W m$^{-2}$. Therefore, the large difference (104 W m$^{-2}$) between the LE values before and after the modification was caused mainly by the differences in the resistance parameterizations. A weak relationship between the residual LE error and $r_s$ values in figures. 9d to 9g further revealed that LE values estimated from the old but simple algorithms in the report by Mu et al. (2007) exhibited less bias (especially when $r_{s,c} < 400$ s m$^{-1}$) because estimation of $g_c$ and $r_{totc}$ became much more complicated in the improved algorithms, as shown in Table 3 (Mu et al., 2011). These results indicate that the uncertainty in ET estimation not only is caused by different resistance estimation methods but also can arise from the same method when using different parameterizations or assumptions (or nonunique parameterizations).

[Figure]

**Figure 8: Differences in surface resistance, $r_s$ (a), and scatter plots showing the effects of $r_s$ on latent heat flux (LE) in terms of residual LE error (b and c).**

**Note: The LE values were estimated using the one-source PM equation and the same inputs except for $r_s$; $r_s$_JA, $r_s$_KP, and $r_s$_PM are $r_s$ values from the method of Jarvis (1976), the method of Katerji and Perrier (1983), and the inverse of the one-source PM equation, respectively; residual LE error = difference between the observed LE and estimated LE.**

[Figure]

**Figure 9: Differences in latent heat flux (LE) estimates from two versions of the PM_Mu algorithm (a) and resistances, $r_{s,c}$ (b) and $r_{s,s}$ (c); scatter plots showing the effects of $r_{s,c}$ ($r_{s,s}$) on LE in terms of residual LE error (d and e) (f and g).**
**Note: The LE values were estimated using the two-source PM equation; $r_{s,c}$ and $r_{s,s}$ represent the surface resistance of vegetation and the soil resistances, respectively; 2007 and 2011 refer to the old algorithms in the report of Mu et al. (2007) and the improved algorithms in the report of Mu et al. (2011); MAE and RMSE represent the mean absolute error and root mean squared error, respectively; residual LE error = difference between the observed LEs and estimated LEs.**

**Comment 3**:

How the resistance models performed under different soil moisture, VPD and radiation conditions? Without a detailed analysis, it would be difficult to assess the scientific value of the paper.

**Responses:**

Thank you.

In fact, the performance of the different PM models at different maize sites was discussed in detail in figure 5 in the previous manuscript. The maize sites exhibited varied plant biophysics (LAI) and soil moisture contents (as well as VPD and net radiation). The results of the statistical analysis revealed that PM_KP performed better than did PM_JA in terms of $R^2$, with a mean value of 0.89 for the former compared with 0.73 for the latter. However, the MAPE values were generally greater than 30% for PM_KP and PM_JA. With respect to PM_Mu, $R^2$ was relatively low, with a mean value of 0.36, and the MAPE values were relatively large, with a mean value of 38.9%. Compared with the one-source PM equation, the two-source PM equation performed relatively poorly, which may likely be due to the two-source PM equation requiring over complicated parameterizations.

Because the first reviewer thought such statistical results of individual flux sites (in the previous figure 5) may not be meaningful for most readers who do not know the sites, we revised the figure by

showing the model performance (MAPE) in the LAI-soil moisture space, attempting to provide intuitive information to readers in the event that they are not familiar with the sites (Fig. 6 as follows). There is no specific pattern in model performance in terms of LAI or soil moisture. In addition, the performance of the PM equation in this study was poor (MAPE > 30%); therefore, we believe it is unnecessary to perform a further analysis.

[Figure]

**Figure 6: Performance of the different models at different maize sites under different leaf area index (LAI) and soil moisture conditions in terms of mean absolute percent error (MAPE): (a) PM_JA method, (b) PM_KP method, (c) PM_Mu method, and (d) the 3T model. Note: the number represents the EC system ID in figure 1c; the data are from figure 4; panels a, b, and c have the same color legend.**

**Comment 4**:

How the 3T model performed under different soil moisture, VPD and radiation conditions?

**Responses:**

Thank you.

In fact, the performance of the 3T model at the different maize sites was discussed in detail in figure 5 of a previous manuscript. The maize sites exhibited varied plant biophysics (LAI) and soil moisture contents. The statistical results revealed that at the maize sites, the mean $R^2$ was 0.88 for the 3T model, with maximum and minimum values of 0.93 and 0.85, respectively; moreover, the MAPE values varied from 13.88% to 22.08%, with a mean value of 17.30%. Please see figure 4 in our reply to comment 3 for details. However, the performance of the 3T model also has no specific pattern. Its performance differed little only when the sites had similar (values close to each other) LAI and soil moisture content conditions, e.g., sites 2 and 12 and sites 5 and 7. For sites 2 and 5 (or 12 and 7), the LAI values were the same, but the MAPE value

for site 5 with a relatively high soil moisture content was smaller than that of site 2 (or the model performed better at site 5). Nonetheless, with an even greater soil moisture content and similar LAI, site 3 was associated with the worst model performance among the three sites.

We also show the performance of the 3T model under different VPD and Rn conditions (Fig. R1 as follows). There is also no specific pattern of the model performance. Thus, it is unnecessary to add the results to the revised version.

[Figure]

**Figure R1: Performance of the 3T model at different maize sites under different VPD and Rn in terms of mean absolute percent error (MAPE). Note: the number represents the EC system ID.**

In accordance with this comment, we added the abovementioned discussion to Section 4.2 in the revised version:

Generally, the 3T model performed much better than did both the one- and two-source PM equations; the MAPE was 19% for the 3T model, whereas it was greater than 32% for the PM equations. The one-source PM equation performed slightly better than that did the two-source PM_Mu. For the one-source method, the PM_JA caused the smallest biases in the LE estimation, whereas the empirical PM_KP model led to the largest uncertainty. Because the study area was a maize-dominated heterogeneous oasis and because the maize fields exhibited varied plant biophysics and soil moisture content conditions (Table 2), we further evaluated the model performance at different maize sites. The mean absolute differences in the LE values between the estimates and the observations were 105, 118, 131, and 60 W m$^{-2}$ for PM_JA, PM_KP, PM_Mu, and the 3T model, respectively. Most of the MAPE values were greater than 30% for the PM equations, whereas they varied from 14% to 22% with a mean value of 19% for the 3T model (Fig. 6). Within the LAI-soil moisture space, the performance of the 3T model exhibited little difference only when flux sites had similar LAI and soil moisture content conditions (values close to each other), such as the values at sites 2 and 12 and sites 5 and 7. For sites 2 and 5 (or 12 and 7), the LAI values were the same, but the MAPE value for site 5 with a higher soil moisture was smaller than that of site 2 (i.e., the model performed better at site 5). Nonetheless, with an even greater soil moisture and similar LAI, site 3 associated with the worst model performance among the three sites. The performance of the 3T model under different VPD and Rn conditions was similar to that in the LAI-soil moisture space (results not shown here). The performance of the models varied for a given study site, with no specific pattern.

**Comment 5**:

A Table of symbols and their unit for different models would greatly improve the readability of the manuscript.

**Responses:**

Thank you.

We will add a table (Table A1) containing symbols and their units for the different models:

Table A1. Summary and description of variables and symbols

| variables and symbols | units | description |
|---|---|---|
| ET | mm | evapotranspiration (evaporation + transpiration) as depth of water |
| $E_c$ | mm | vegetation transpiration |
| $E_s$ | mm | soil evaporation |
| $L$ | $J\,kg^{-1}\,K^{-1}$ | latent heat of vaporization |
| $L(ET)$ | $W\,m^{-2}$ | evapotranspiration (evaporation + transpiration) as latent heat flux |
| $LE_c$ | $W\,m^{-2}$ | vegetation transpiration as latent heat flux |
| $LE_s$ | $W\,m^{-2}$ | soil evaporation as latent heat flux |
| $R_n$ | $W\,m^{-2}$ | net radiation |
| $R_{n,s}$ | $W\,m^{-2}$ | net radiation of the soil component |
| $R_{n,c}$ | $W\,m^{-2}$ | net radiation of the vegetation component |
| $R_s$ | $W\,m^{-2}$ | solar radiation |
| $G$ | $W\,m^{-2}$ | soil heat flux |
| $H$ | $W\,m^{-2}$ | sensible heat flux |
| $T_a$ | °C | air temperature |
| $T_{0s}$ | °C | soil surface temperature |
| $T_{0c}$ | °C | canopy temperature |
| $T_L$ | °C | lower air temperature limits of stomatal activity method |
| $T_{op}$ | °C | optimal air temperature limits of stomatal activity method |
| $T_H$ | °C | upper air temperature limits of stomatal activity method |
| $r_a$ | $s\,m^{-1}$ | aerodynamic resistance |
| $r_{a,c}$ | $s\,m^{-1}$ | aerodynamic resistance for canopy |
| $r_{a,s}$ | $s\,m^{-1}$ | aerodynamic resistance for soil surface |

| | | |
|---|---|---|
| $r_s$ | s m$^{-1}$ | surface resistance |
| $r_{smin}$ | s m$^{-1}$ | minimum stomatal resistance under optimal conditions |
| $r_{s,c}$ | s m$^{-1}$ | canopy resistance |
| $r_{s,s}$ | s m$^{-1}$ | soil surface resistance |
| $g_s$ | m s$^{-1}$ | leaf stomatal (surface) conductance |
| $g_c$ | m s$^{-1}$ | canopy conductance |
| $\gamma$ | kPa °C$^{-1}$ | psychrometric constant |
| $VPD$ | kPa | vapor pressure deficit |
| $\Delta$ | − | slope of the saturation vapor pressure with respect to temperature |
| $Rh$ | − | relative humidity |
| $f_c$ | − | fractional vegetation cover |
| $\rho_a$ | kg m$^{-3}$ | mean air density at constant pressure |
| $C_p$ | MJ kg$^{-1}$ K$^{-1}$ | specific heat of the air |
| $z_r$ | m | reference height |
| $z_{0m}$ | m | surface roughness length for momentum |
| $z_{0h}$ | m | surface roughness length for heat |
| $u_{zr}$ | m s$^{-1}$ | wind speed at reference height |
| $d$ | m | displacement height |
| $\theta$ | cm$^3$ cm$^{-3}$ | soil water |
| $\theta_w$ | cm$^3$ cm$^{-3}$ | wilting point |
| $\theta_f$ | cm$^3$ cm$^{-3}$ | field capacity |

**Comment 6**:

Analysis of Sensible heat fluxes should also be included in a condensed manner.

**Responses:**

Thank you.

In fact, we did not estimate the sensible heat flux (H). When estimating latent heat flux (LE) via residual methods of the energy-balance equation, H should be estimated first. However, in this study, we did not use a residual method of the energy-balance equation. We have H data from the tower observation, but we believe it may be strange to analyze H when discussing uncertainty in LE estimation without parameterizing H.

**Comment 7**:

How 3T model avoids the parameterization of the resistances? This is a good side of the model. However, it needs to be described in a condensed manner.

**Responses:**

Thank you.

We believe a clear description of the 3T model, such as nonparameterization of resistances, is necessary. Nonetheless, we have attempted to shorten the description while keeping key information:

2.3 The 3T model

The 3T model, which is derived from the energy balance equation, was first developed by Qiu (1996) and is loosely related to the three-leaf model of Paw U and Daughtry (1984). A unique characteristic of the 3T model is that the estimation of ET does not explicitly include any resistance parameterizations. A reference surface temperature (a dry surface without evaporation or transpiration) is used to eliminate latent heat and the surface resistance to water vapor under the assumption that the $r_a$ for the reference soil is the same as that for other soil surfaces, resulting in Eq. (16) (see details in Qiu et al., 1996, 1998):

$$LE_s = R_{n,s} - G_s - \left(R_{n,sr} - G_{sr}\right)\frac{T_{0s} - T_a}{T_{0sr} - T_a} \qquad\qquad \text{soil} \qquad\qquad (16)$$

where $T_{0s}$ in K is the temperature of the soil component; other variables are similar to previous ones, with subscripts "s" and "sr" indicating the soil component and the reference dry soil, respectively.

A similar technique was used to obtain Eq. (17) by introducing a reference dry vegetation surface:

$$LE_c = R_{n,c} - R_{n,cr}\frac{T_{0c} - T_a}{T_{0cr} - T_a} \qquad\qquad \text{vegetation} \qquad\qquad (17)$$

where $T_{0c}$ is the temperature of the vegetation component and subscripts "c" and "cr" indicate the vegetation component and the reference dry canopy, respectively.

The total latent heat flux equation can then be calculated using Eq. (7). This unique feature makes the 3T model a relatively simple method for RS applications (Xiong & Qiu, 2011, 2014; Xiong et al., 2015; Wang Y. et al., 2016).